# Q-Delta: Beyond Key–Value Associative State Evolution

**Sumin Park** [1]  **Seojin Kim** [1]  **Noseong Park** [1]

## Abstract

Linear attention reformulates sequence modeling as recurrent state evolution, enabling efficient linear-time inference. Under the key–value associative paradigm, existing approaches restrict the role of the query to the readout operation, decoupling it from state evolution. We show that query-conditioned state readout induces a structured value prediction over accumulated memory that complements key-based retrieval. Based on this insight, we propose Q-Delta, a query-aware delta rule that integrates mixed key–query prediction errors into state evolution, enabling jointly corrective dynamics while preserving delta-rule efficiency. We establish stability guarantees for the resulting dynamics and derive a hardware-efficient chunkwise-parallel formulation with a custom Triton implementation. Empirical results demonstrate stable optimization, competitive throughput, and consistent improvements over strong baselines on language modeling and long-context retrieval tasks. Code is available at https://github.com/psmiz/Q-Delta.

## 1. Introduction

The Transformer architecture achieves strong sequence modeling performance with its softmax-based self-attention mechanism (Vaswani et al., 2023), but incurs quadratic time and memory complexity with respect to sequence length. This limitation has motivated a line of work on linear Transformers, which replace softmax attention with kernelized or algebraically decomposable feature mappings $\phi(\cdot)$ that allow the attention computation to be algebraically reordered as $\phi(Q)\big(\phi(K)^\top V\big)$, enabling linear-time inference and training scalability (Chevalier, 2018; Wang et al., 2020; Katharopoulos et al., 2020). This factorization enables an online realization in which the term $\phi(K)^\top V$ is maintained

as an incrementally updated state, $S_t = \sum_{i=1}^{t} v_i \phi(k_i)^\top$, revealing the attention as recurrent state evolution where information is written into a memory state by key–value outer products and retrieved via a query readout, $o_t = S_t \phi(q_t)$.

This perspective leads to a unifying interpretation of linear attention as querying an evolving key–value associative memory. Rather than modeling explicit pairwise interactions between tokens, linear attention emphasizes how information is incrementally written, stored, and retrieved from a shared memory structure through iterative state updates. Under this view, a range of prior works, including kernelized linear attention (Kitaev et al., 2020; Wang et al., 2020; Sun et al., 2023; Yang et al., 2024; Sun et al., 2024) and selective state space models (SSMs) (Gu et al., 2020; Smith et al., 2023; Gu & Dao, 2024; Dao & Gu, 2024), can be viewed as linear RNN–style architectures that replace explicit attention maps with structured state evolution with recurrent update rules.

Purely additive updates, however, lack mechanisms for adaptive memory modifications, failing to selectively revise or remove previously stored information. This results in increased key collisions and degraded retrieval accuracy as sequence length grows (Schlag et al., 2021). Delta-rule–based updates (Liu et al., 2024; Yang et al., 2025b;a) address this limitation by refining the state in response to retrieval error, the discrepancy between the observed value and the value retrieved by the current key. Longhorn (Liu et al., 2024) reveals that this error-driven update reduces to an online regression step on the key–value prediction objective, enabling selective modification of the recurrent state while preserving linear-time recurrence (Liu et al., 2024). Recent linear transformers and SSMs adopt this perspective to reinterpret recurrent state evolution as amortized online learning, providing a principled foundation for improved in-context retrieval and memory control (Brown et al., 2020; Olsson et al., 2022).

Despite these advances, existing linear RNN models share a common structural assumption: state evolution is governed primarily by key–value interactions, while the query is used only to read out the evolved state. While this separation follows naturally from the original attention formulation, it implicitly assumes that query plays no informative role in shaping state dynamics. We question this conventional view

---

[1]Korea Advanced Institute of Science and Technology (KAIST), Daejeon, Republic of Korea. Correspondence to: Noseong Park <noseong@kaist.ac.kr>.

*Proceedings of the 43rd International Conference on Machine Learning*, Seoul, South Korea. PMLR 306, 2026. Copyright 2026 by the author(s).

of query as a passive readout mechanism by re-examining the role of query-based readout in recurrent state update process. In this work, we show that querying the state yields a value prediction that reflects information stored across the accumulated memory trace, providing a distinct but complementary signal to key-based retrieval.

Motivated by this observation, we introduce **Q-Delta**, a query-aware delta rule that enables predictive state evolution by incorporating query-conditioned feedback directly into recurrent memory updates. Q-Delta jointly considers a key-retrieved value estimate $\hat{v}_t = S_{t-1} k_t$ and a query-conditioned value prediction $\hat{o}_t = S_{t-1} q_t$, and updates the memory using a mixed correction signal that couples these complementary value estimators. We show that the resulting dynamics remain stable and satisfy global geometric error contraction under mild empirical conditions, and we further derive a chunkwise-parallel formulation compatible with hardware-efficient training implemented in Triton kernel.

Our main contributions are summarized as follows:

- We revisit the role of query readout in linear attention, showing that it induces a structured value prediction over accumulated memory.

- We propose Q-Delta, a query-aware delta rule that integrates mixed key–query prediction errors into state evolution, together with a hardware-efficient chunkwise-parallel Triton implementation.

- We establish a stability theory for Q-Delta, proving one-step contraction and global stability of the mixed key–query error under empirical alignment conditions.

- Empirically, Q-Delta consistently outperforms prior linear Transformers and SSMs baselines on language modeling and long-context retrieval tasks.

## 2. Backgrounds

### 2.1. Linear Transformers

Recent work (Katharopoulos et al., 2020; Sun et al., 2023; Yang et al., 2024; Liu et al., 2024; Gu & Dao, 2024; Yang et al., 2025b;a) has shown that linear attention can be equivalently formulated as a linear recurrent model with a matrix-valued state. In its classic form, omitting normalization and feature activations, linear attention admits the recurrence

$$S_t = S_{t-1} + v_t k_t^\top \in \mathbb{R}^{d_v \times d_k}, \quad o_t = S_t q_t \in \mathbb{R}^{d_v}, \quad (1)$$

where $d_k$ and $d_v$ represent the (head) dimensions for $q_t, k_t \in \mathbb{R}^{d_k}$ and $v_t \in \mathbb{R}^{d_v}$ and $S_t$ accumulates rank one key–value outer products over time. Longhorn reframes this update rule as an online learning, interpreting the state update as the implicit solution of an online regression problem that learns a linear map from keys to values. (Olsson et al., 2022; Liu

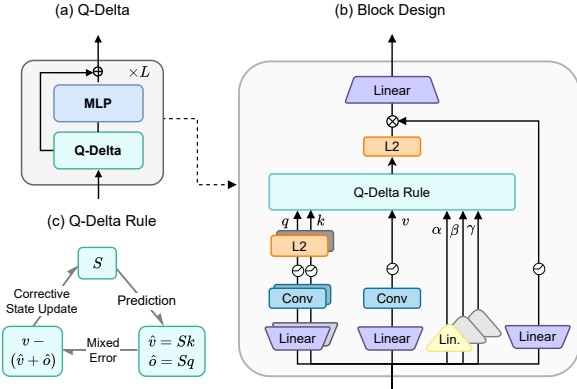

*Figure 1.* **Architecture overview and block design of Q-Delta.** (a) Q-Delta module within a Transformer block. (b) Block-level implementation illustrating how queries, keys, and values are projected and combined with gating signals. (c) The Q-Delta update rule, where the recurrent state produces a key-retrieved value $\hat{v} = Sk$ and a query-conditioned prediction $\hat{o} = Sq$, which are then combined into a mixed error for corrective state evolution.

et al., 2024) Under this view, designing a linear sequence-mixing model reduces to specifying an online loss and a regularizer that govern how new key–value information is incorporated into state (Sun et al., 2025; Yang et al., 2025b; 2024; Hu et al., 2025). This perspective provides a unified framework for understanding linear Transformers and their extensions as online linear regressors.

**Delta-rule and gated extensions.** While the additive update in Eq. (1) is efficient, it lacks a mechanism for selectively overwriting or correcting stored key-value associations. Delta-based models (Liu et al., 2024; Yang et al., 2025b) address this limitation by modifying the state along the direction of the current key. The Deltanet (Yang et al., 2025b) updates the state as

$$S_t = S_{t-1}(I - \beta_t k_t k_t^\top) + \beta_t v_t k_t^\top, \quad (2)$$

where $\beta_t \in (0, 1)$ controls the writing strength, dynamically erasing the old value $v_t^{\text{old}} = S_{t-1} k_t$ retrieved by $k_t$ and writing a new one $v_t^{\text{new}} = v_t$. GatedDeltaNet (Yang et al., 2025a) further augments this update with multiplicative gating, yielding recurrences in the form

$$S_t = S_{t-1}\big(\alpha_t(I - \beta_t k_t k_t^\top)\big) + \beta_t v_t k_t^\top, \quad (3)$$

where $\alpha_t$ controls the state decay. Closely related decay-based formulations also arise in Mamba2 (Gu & Dao, 2024; Dao & Gu, 2024), whose SSM dynamics can be expressed as a linear recurrence with a decay term.

**Explicit memory update via online regression.** Beyond implicit memory encoded in recurrent state updates, a growing line of work treats memory as an explicit module that is

*Table 1.* Comparison of linear RNN models and their online learning objectives under the framework of Liu et al. (2024).

| Method | Online Learning Objective | Online Update |
|---|---|---|
| LA | $\|S_t - S_{t-1}\|_F^2 - 2\langle S_t k_t,\, v_t\rangle$ | $S_t = S_{t-1} + v_t k_t^\top$ |
| Mamba2 | $\|S_t - \alpha_t S_{t-1}\|_F^2 - 2\langle S_t k_t,\, v_t\rangle$ | $S_t = \alpha_t S_{t-1} + v_t k_t^\top$ |
| Longhorn | $\|S_t - S_{t-1}\|_F^2 - \beta_t \|S_t k_t - v_t\|_2^2$ | $S_t = S_{t-1}\left(I - \epsilon_t k_t k_t^\top\right) + \epsilon_t v_t k_t^\top, \quad \epsilon_t = \dfrac{\beta_t}{1 + \beta_t\, k_t^\top k_t}$ |
| DeltaNet | $\|S_t - S_{t-1}\|_F^2 - 2\langle S_t k_t,\, \beta_t(v_t - S_{t-1}k_t)\rangle$ | $S_t = S_{t-1}\left(I - \beta_t k_t k_t^\top\right) + \beta_t v_t k_t^\top$ |
| GatedDeltaNet | $\|S_t - \alpha_t S_{t-1}\|_F^2 - 2\langle S_t k_t,\, \beta_t(v_t - \alpha_t S_{t-1}k_t)\rangle$ | $S_t = S_{t-1}\left(\alpha_t\left(I - \beta_t k_t k_t^\top\right)\right) + \beta_t v_t k_t^\top$ |
| **Q-Delta (ours)** | $\|S_t - \alpha_t S_{t-1}\|_F^2 - 2\big\langle S_t k_t,\, \beta_t(v_t - \alpha_t S_{t-1}k_t - \lambda_t\, \alpha_t S_{t-1} q_t)\big\rangle$ | $S_t = S_{t-1}\left(\alpha_t\left(I - \beta_t\left(k_t k_t^\top + \lambda_t\, q_t k_t^\top\right)\right)\right) + \beta_t v_t k_t^\top$ |

continuously updated by online learning rules at inference time. Test-Time Training (TTT) (Sun et al., 2025) optimizes the state via online gradient descent on a key-value prediction loss during both training and inference,

$$S_t = S_{t-B} - \sum_{i=1}^{B} \eta_i \nabla_S \big\| Sk_i - v_i \big\|^2. \qquad (4)$$

Similarly, Titans (Behrouz et al., 2024) introduce a neural long-term memory module whose parameters are updated at test time to memorize key–value associations, with decay and momentum controlling forgetting and retention.

**Chunkwise Parallel Form.** Although linear recurrences achieve an efficient linear-complexity with $\mathcal{O}(LD^2)$, their fully sequential nature limits training efficiency on modern hardware that favors parallelized computations. To address this, recent works reformulate linear recurrences in a chunkwise parallel manner, combining inter-chunk recurrence with intra-chunk parallel computation. The key idea is to partition the sequence into contiguous chunks of length $C$, allowing parallel computation within each chunk while maintaining a recurrent dependency across chunks. For the basic linear attention, the chunkwise formulation is

$$\begin{aligned} S_{[t+1]} &= S_{[t]} + V_{[t]}^\top K_{[t]}, \\ O_{[t]} &= Q_{[t]} S_{[t]}^\top + \big(Q_{[t]} K_{[t]}^\top \odot \mathrm{M}_{[t]}\big) V_{[t]}, \end{aligned} \qquad (5)$$

where $K_{[t]}, Q_{[t]}, V_{[t]} \in \mathbb{R}^{C \times D}$ stack the key, query, and value vectors within the chunk, and $\mathrm{M}_{[t]} \in \mathbb{R}^{C \times C}$ enforces causality within the chunk. More structured delta-rule recurrence can be expressed as

$$\begin{aligned} S_{[t+1]} &= S_{[t]} + \big(U_{[t]} - W_{[t]} S_{[t]}\big) K_{[t]}, \\ O_{[t]} &= Q_{[t]} S_{[t]}^\top + \big(Q_{[t]} K_{[t]}^\top \odot M\big)\big(U_{[t]} - W_{[t]} S_{[t]}\big). \end{aligned} \qquad (6)$$

Here $U_{[t]}$ and $W_{[t]}$ are chunkwise matrices induced by the UT transform to ensure sequential delta update in chunk-level recurrence (Joffrain et al., 2006; Dominguez & Orti, 2018; Yang et al., 2025b).

$$\begin{aligned} \mathbf{T}_{[t]} &= \Big(\mathbf{I} + \mathrm{tril}\Big(\mathrm{diag}(\boldsymbol{\beta}_{[t]})\, \mathbf{K}_{[t]} \mathbf{K}_{[t]}^\top, -1\Big)\Big)^{-1} \mathrm{diag}(\boldsymbol{\beta}_{[t]}), \\ \mathbf{W}_{[t]} &= \mathbf{T}_{[t]} \mathbf{K}_{[t]}, \quad \mathbf{U}_{[t]} = \mathbf{T}_{[t]} \mathbf{V}_{[t]}. \end{aligned} \qquad (7)$$

This formulation preserves the original delta-rule dynamics while enabling efficient hardware-parallelism.

## 3. State Beyond Key–Value Association

Across existing linear attention and SSMs, the state is predominantly interpreted as a key–value associative memory, while the query $q_t$ is used exclusively at readout time. Under this interpretation, the query serves only as a passive readout mechanism and plays no role in shaping the state dynamics. In this section, we question this assumption and show that the query readout itself encodes structured value information derived from the state, motivating a refined view of state evolution.

### 3.1. Query for Value Prediction

Prior work has shown that query-induced state readout, $o_t = S_t q_t$, following the linear recurrences can be expressed as a value-weighted aggregation of past tokens (Yang et al., 2025a). We extend this characterization to the readout taken from the prior state, $\hat{o}_t := S_{t-1} q_t$, and generalize it to an arbitrary linear transition operator, so that the temporally mixed value-aggregation form holds uniformly across generic linear recurrence rules, including delta-rule and gated recurrences. We use this reformulation to motivate query-conditioned state evolution in the following sections.

**Query readout as temporally mixed value.** Consider a generic form of recurrent state sequence $\{S_t\}_{t \geq 1}$ defined as

$$S_t = S_{t-1} P_t + \eta_t v_t k_t^\top, \qquad S_0 = 0, \qquad (8)$$

where $v_t \in \mathbb{R}^{d_v}$, $k_t \in \mathbb{R}^{d_k}$, $\eta_t \in \mathbb{R}$, and $P_t \in \mathbb{R}^{d_k \times d_k}$ is a linear state transition operator. Given this recurrence, state for each $t$ can be written as a linear combination of previously written value vectors

$$S_{t-1} = \sum_{\tau=1}^{t-1} v_\tau\, b_{\tau,t-1}^\top, \qquad (9)$$

where the coefficient vectors $\{b_{\tau,t-1}\}_{\tau<t} \subset \mathbb{R}^{d_k}$ satisfy the backward recursion as

$$b_{\tau,t} = P_t^\top b_{\tau,t-1} \quad (\tau < t), \qquad b_{t,t} = \eta_t k_t. \qquad (10)$$

Unrolling this gives the closed form, for any $\tau < t$,

$$b_{\tau,t-1} = \eta_\tau \left( \prod_{j=\tau+1}^{t-1} P_j^\top \right) k_\tau. \tag{11}$$

Consequently, the query-conditioned prediction from the prior state, $S_{t-1}q_t$, admits the temporally mixed value form as follows

$$\hat{o}_t = \sum_{\tau=1}^{t-1} \gamma_{\tau,t} v_\tau, \qquad \gamma_{\tau,t} := b_{\tau,t-1}^\top q_t \in \mathbb{R}, \tag{12}$$

so the query readout lies in the span of previously stored values and acts as a weighted value aggregation over past timesteps (see Appendix C.1 for derivations). This result specializes to a standard linear attention ($P_t = I$) (Katharopoulos et al., 2020), the delta rule ($P_t = I - \beta_t k_t k_t^\top$) (Yang et al., 2025b), and gated delta variants ($P_t = \alpha_t(I - \beta_t k_t k_t^\top)$) (Yang et al., 2025a).

**Attention over accumulated memory.** Define the time-evolved key $\tilde{k}_{\tau,t}$ associated with past key–value pair $(k_\tau, v_\tau)$ at query time $t$ as

$$\tilde{k}_{\tau,t} := \left( \prod_{j=\tau+1}^{t-1} P_j^\top \right) k_\tau \quad \in \mathbb{R}^{d_k}, \tag{13}$$

which then gives $b_{\tau,t-1} = \eta_\tau \tilde{k}_{\tau,t}$. Intuitively, $\tilde{k}_{\tau,t}$ encodes how the original key $k_\tau$ is transformed by subsequent state transitions via $\{P_j\}_{j=\tau+1}^{t-1}$. It determines how strongly the current query $q_t$ can retrieve the value $v_\tau$ from the accumulated memory at current time $t$.

Using the definition of the time-evolved key $\tilde{k}_{\tau,t}$, the query-induced prediction formed from the prior state, $\hat{o}_t := S_{t-1}q_t$, can be written as

$$\hat{o}_t = \sum_{\tau=1}^{t-1} \gamma_{\tau,t} v_\tau, \qquad \gamma_{\tau,t} = \eta_\tau q_t^\top \tilde{k}_{\tau,t}. \tag{14}$$

Equivalently,

$$\hat{o}_t = \sum_{\tau=1}^{t-1} \langle q_t, \tilde{k}_{\tau,t} \rangle_{\eta_\tau} v_\tau. \tag{15}$$

This has the form of unnormalized attention, in which the current query $q_t$ is matched against the time-evolved keys $\{\tilde{k}_{\tau,t}\}_{\tau < t}$ to mix values stored across time. Thus, $\hat{o}_t$ is a query-dependent value prediction obtained by attending to the entire accumulated key–value memory, with the state transition operators $\{P_j\}_{j=\tau+1}^{t-1}$ governing how past keys are reshaped over time.

**Why query readout matters.** The analysis above shows that the query-conditioned readout $\hat{o}_t = S_{t-1}q_t$ is a structured value aggregation driven by attending over the accumulated memory with time-evolved keys. This aggregation

lies in the same value space as the key readout $S_{t-1}k_t$, but is weighted by attention-like similarities between the current query and past keys, rather than key–key self-similarity. As a result, $\hat{o}_t$ gives a query-induced value information that is already encoded in the state, which is not accessible through key-based recall alone.

This value-aggregation form is, on its own, an algebraic identity that holds for any probe vector. What distinguishes the query is its role in the recurrence, $q_t$ is the direction along which the state is finally read out, since the layer output is $o_t = S_t q_t$. The query-conditioned prediction $\hat{o}_t = S_{t-1}q_t$ is not an arbitrary projection of the state, but the model's own value prediction along the very direction through which the memory is ultimately consumed downstream. Yet, conventional delta-rule corrects the state only against the key-retrieved value $\hat{v}_t = S_{t-1}k_t$. The query readout $\hat{o}_t$ adds a complementary corrective signal to state-evolution process, aligning it with the direction the state is actually read out, motivating its inclusion in the update.

### 3.2. Complementary Error Signals

**Mixed prediction errors.** Given keys $k_t \in \mathbb{R}^{d_k}$, values $v_t \in \mathbb{R}^{d_v}$, queries $q_t \in \mathbb{R}^{d_k}$, and the prior state $S_{t-1} \in \mathbb{R}^{d_v \times d_k}$, we first compute two distinct value estimates,

$$\hat{v}_t := S_{t-1}k_t, \qquad \hat{o}_t := S_{t-1}q_t. \tag{16}$$

Here, $\hat{v}_t$ corresponds to the key-retrieved value, while $\hat{o}_t$ is a query-conditioned prediction.

Intuitively, the two estimates $\hat{v}_t$ and $\hat{o}_t$ correspond to distinct projections of the same accumulated memory state. The key-based readout $\hat{v}_t = S_{t-1}k_t$ evaluates the memory along the direction of the current key, yielding the value currently associated with $k_t$ under the learned key–value mapping. In contrast, $\hat{o}_t = S_{t-1}q_t$ evaluates the same state along a different direction specified by the query, producing a value aggregation shaped by how the query aligns with time-evolved keys in memory. Although both readouts are formed by aggregating past values, they generally induce different weightings over those values. This distinction indicates that $\hat{v}_t$ and $\hat{o}_t$ encode complementary information present in the state, revealed by different projections. Therefore, incorporating both in the state update process corrects this mixed prediction error, enabling more informative memory updates.

Figure 2 provides empirical evidence that the key-retrieved value $\hat{v}_t = S_{t-1}k_t$ and the query-conditioned prediction $\hat{o}_t = S_{t-1}q_t$ encode complementary, rather than redundant, information from the recurrent state. Left: we plot the distribution of cosine similarities $\langle \hat{o}_t, \hat{v}_t \rangle / (\|\hat{o}_t\| \|\hat{v}_t\|)$ gathered across 10000 timesteps and 3 layers (5, 10, 15) of 340M Q-Delta. The distribution is centered close to zero (mean $\approx 0.07$), indicating that $\hat{o}_t$ and $\hat{v}_t$ occupy largely decorre-

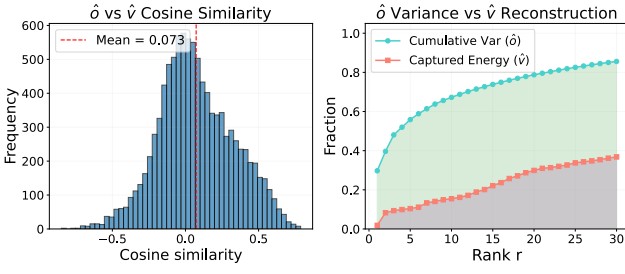

*Figure 2.* Complementarity analysis between $\hat{v}$ and $\hat{o}$. Left: Distribution of cosine similarity between $\hat{v}$ and $\hat{o}$, showing low alignment on average. Right: Cumulative $\hat{o}$ variance and $\hat{v}$ reconstruction energy across principal subspace rank $r$ of $\hat{o}$.

lated directions in value space, despite being derived from the same state $S_{t-1}$. Right: we analyze complementarity at the subspace level by comparing the cumulative variance explained by the principal components of $\hat{o}$ with the fraction of $\hat{v}$ energy captured when projected onto the corresponding subspace of $\hat{o}$. Specifically, we perform PCA on samples of $\hat{o}$ and measure the reconstruction energy of $\hat{v}$ under the top-$r$ principal subspace. The substantial gap between the two curves shows that directions accounting for most of the variance of $\hat{o}$ explain only a limited portion of the energy of $\hat{v}$. Together, these results indicate that query-based predictions emphasize value components that are not well represented by key-based recall alone, supporting the use of mixed errors, $\hat{v}_t$ and $\hat{o}_t$, as complementary error signals for the state evolution under Q-Delta update rule.

### 3.3. Q-Delta: Query-Aware Delta Rule

We now introduce Q-Delta, a query-aware extension of the delta rule that incorporates query-conditioned prediction feedback into state evolution. Q-Delta builds upon delta-based associative memory updates, while allowing both key and query to participate in correcting the stored state. Table 1 summarizes Q-Delta in comparison to prior linear RNN models under a unified online learning objective framework and Figure 1 illustrates the mechanism of Q-Delta.

#### 3.3.1. SEQUENTIAL RECURRENCE.

**Q-Delta rule**  We propose Q-Delta, a query-aware delta rule:

$$S_t = S_{t-1} + \beta_t\big(v_t - \hat{v}_t - \lambda_t\hat{o}_t\big)k_t^\top, \qquad (17)$$

where $\beta_t \in [0, 1]$ controls the update strength and $\lambda_t \in [0, 1]$ modulates the influence of query-based feedback.

Rewriting Eq. (17) yields an equivalent linear form,

$$S_t = S_{t-1}\big(I - \beta_t(k_tk_t^\top + \lambda_t q_tk_t^\top)\big) + \beta_t v_tk_t^\top. \quad (18)$$

Including a forget gate $\alpha_t \in (0, 1)$, the final Q-Delta update

rule is as follows:

$$S_t = \alpha_t S_{t-1}\big(I - \beta_t(k_tk_t^\top + \lambda_t q_tk_t^\top)\big) + \beta_t v_tk_t^\top. \quad (19)$$
$$o_t = S_tq_t.$$

#### 3.3.2. CHUNKWISE PARALLEL FORM.

We now derive a hardware-efficient chunkwise-parallel formulation for Q-Delta referring to the chunkwise expansion strategy of GatedDeltaNet. Defining $x_t := k_t + \lambda_t q_t$, the Q-Delta recurrence follows:

$$S_t = \alpha_t S_{t-1}\big(I - \beta_t x_tk_t^\top\big) + \beta_t v_tk_t^\top, \qquad (20)$$

where $\lambda_t \in (0, 1)$ is a learnable head-wise query-feedback coefficient (see Appendix B for parameterization). Fix a chunk indexed by $[t]$ consisting of $C$ consecutive timesteps $\{t_1, \ldots, t_C\}$, and denote the chunk entrance state by $S_{[t]} := S_{[t]}^0 = S_{[t-1]}^C$. For each timestep $t_i$, define $P_{t_i} := I - \beta_{t_i}x_{t_i}k_{t_i}^\top$. By partially expanding the recurrence, the state after $r \le C$ steps within the same chunk can be written as

$$S_{[t]}^r = S_{[t]}\underbrace{\Big(\prod_{i=1}^r \alpha_{t_i}P_{t_i}\Big)}_{=:F_{[t]}^r} + \underbrace{\sum_{i=1}^r\Big(\beta_{t_i}v_{t_i}k_{t_i}^\top\prod_{j=i+1}^r \alpha_{t_j}P_{t_j}\Big)}_{=:G_{[t]}^r}.$$
$$(21)$$

Let $\gamma_{[t]}^r := \prod_{i=1}^r \alpha_{t_i}$. Then $F_{[t]}^r = \gamma_{[t]}^r P_{[t]}^r$, where

$$P_{[t]}^r := \prod_{i=1}^r P_{t_i} = \prod_{i=1}^r (I - \beta_{t_i}x_{t_i}k_{t_i}^\top). \qquad (22)$$

Following the extended WY representation (Bischof & Loan, 1985) from GatedDeltaNet, there exist vectors $w_{[t]}^i \in \mathbb{R}^{d_k}$ and $u_{[t]}^i \in \mathbb{R}^{d_v}$ defined as

$$w_{[t]}^r = \beta_{t_r}\left(x_{t_r} - \sum_{i=1}^{r-1} w_{[t]}^i\big(k_{t_i}^\top x_{t_r}\big)\right),$$
$$\tilde{u}_{[t]}^r = \beta_{t_r}\left(v_{t_r} - \sum_{i=1}^{r-1}\frac{\gamma_{[t]}^r}{\gamma_{[t]}^i}\tilde{u}_{[t]}^i\big(k_{t_i}^\top x_{t_r}\big)\right),$$
$$(23)$$

such that (derivations in D.1)

$$P_{[t]}^r = I - \sum_{i=1}^r w_{[t]}^ik_{t_i}^\top, \quad G_{[t]}^r = \sum_{i=1}^r\frac{\gamma_{[t]}^r}{\gamma_{[t]}^i}\tilde{u}_{[t]}^ik_{t_i}^\top. \quad (24)$$

Substituting the WY forms into Eq. (21) gives

$$\begin{aligned}
S_{[t]}^r &= \gamma_{[t]}^r S_{[t]}P_{[t]}^r + G_{[t]}^r \\
&= \gamma_{[t]}^r S_{[t]}\Big(I - \sum_{i=1}^r w_{[t]}^ik_{t_i}^\top\Big) + \sum_{i=1}^r\frac{\gamma_{[t]}^r}{\gamma_{[t]}^i}\tilde{u}_{[t]}^ik_{t_i}^\top \\
&= \gamma_{[t]}^r S_{[t]} + \sum_{i=1}^r\big(\tilde{u}_{[t]}^i - \gamma_{[t]}^i S_{[t]}w_{[t]}^i\big)\frac{\gamma_{[t]}^r}{\gamma_{[t]}^i}k_{t_i}^\top. \quad (25)
\end{aligned}$$

At the end of the chunk ($r = C$), define the scaled state $\overrightarrow{S}_{[t]} := \gamma_{[t]}^C S_{[t]}$, $\overleftarrow{W}_{[t]} := [\gamma_{[t]}^1 w_{[t]}^1, \dots, \gamma_{[t]}^C w_{[t]}^C]$, $Q_{[t]} = [q_{t_1}, \dots, q_{t_C}]$, and $\overrightarrow{K}_{[t]} := (\Gamma_{[t]})_{C(\cdot)} K_{[t]}$ where $(\Gamma_{[t]})_{ij} = \frac{\gamma_{[t]}^i}{\gamma_{[t]}^j}$. Then chunk-level state update admits the compact form

$$S_{[t+1]} = \overrightarrow{S}_{[t]} + \left(\widetilde{U}_{[t]} - \overleftarrow{W}_{[t]} S_{[t]}\right)^\top \overrightarrow{K}_{[t]}$$
$$O_{[t]} = \overleftarrow{Q}_{[t]} S_{[t]}^\top + \left(Q_{[t]} K_{[t]}^\top \odot M\right)\left(\widetilde{U}_{[t]} - \overleftarrow{W}_{[t]} S_{[t]}\right) \tag{26}$$

such that $S_{[t+1]} \in \mathbb{R}^{d_v \times d_k}$ and $O_{[t]} \in \mathbb{R}^{C \times d_v}$ and $M$ is the causal mask. The vectors $\widetilde{U}_{[t]}$ and $W_{[t]}$ can be computed efficiently using a UT transform referring to DeltaNet, yielding a hardware-efficient chunkwise-parallel algorithm for Q-Delta (derivations in Appendix D.2):

$$\widetilde{U}_{[t]} = \left[I + \text{Lower}\big(\text{d}(\beta_{[t]})(\Gamma_{[t]} \odot X_{[t]} K_{[t]}^\top)\big)\right]^{-1} \text{d}(\beta_{[t]}) V_{[t]},$$
$$W_{[t]} = \left[I + \text{Lower}\big(\text{d}(\beta_{[t]}) X_{[t]} K_{[t]}^\top\big)\right]^{-1} \text{d}(\beta_{[t]}) X_{[t]}. \tag{27}$$

We also provide a Triton (Tillet et al., 2019) kernel specific to both fully recurrent and chunkwise parallelized Q-Delta.

### 3.3.3. STABILITY ANALYSIS OF Q-DELTA DYNAMICS

Here we analyze the prediction error dynamics of the proposed Q-Delta update rule. While Q-Delta is motivated by correcting a mixed prediction error involving both key- and query-induced memory readouts, its update rule does not correspond to a strict gradient descent step on $\|v_t - S_{t-1}(k_t + \lambda q_t)\|^2$ unlike other delta rules under standard key-value association paradigm. We therefore provide a theoretical analysis of the stability and error contraction properties induced by this recurrence, showing that key–query jointly corrective feedback leads to controlled error dynamics under mild empirical conditions, despite not under a strict gradient descent interpretation.

**Lemma 3.1** (One-step contraction of mixed prediction error under Q-Delta). *Let $k_t, q_t \in \mathbb{R}^d$ and $\lambda_t \in [0, 1]$, and define the mixed input $x_t := k_t + \lambda_t q_t$. Consider the Q-Delta update*

$$S_t = S_{t-1} + \beta_t(v_t - S_{t-1}x_t)k_t^\top, \qquad \beta_t \in (0, 1].$$

*Assume that the scalar alignment $a_t := k_t^\top x_t$ satisfies $\beta_t a_t \in (0, 2)$ almost surely, and define*

$$\rho := \sup_t |1 - \beta_t a_t| \in (0, 1).$$

*Then the mixed prediction error contracts in one step:*

$$\|v_t - S_t x_t\| \le \rho \|v_t - S_{t-1}x_t\| \quad \text{almost surely for all } t.$$

Lemma 3.1 establishes a sufficient condition under which the mixed prediction error strictly decreases under a single-step Q-Delta update. However, in practice, both $\beta_t$ and $a_t$

are data-dependent and vary across timesteps, so a single analytic bound on $\beta_t a_t$ cannot be determined. Nevertheless, empirical measures show that $\beta_t a_t$ consistently stays within the contraction regime during training. Figure 3-(a) shows the distribution of $\beta_t a_t$ collected from the full training steps on 15B tokens across all layers of a 340M Q-Delta model, where values are tightly concentrated within the range of contraction $\beta_t a_t \in (0, 2)$ with mean 0.043.

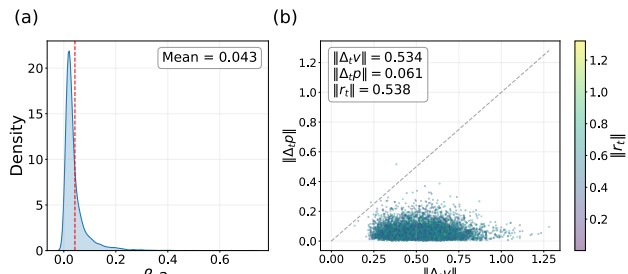

*Figure 3.* Empirical stability analyses of Q-Delta dynamics. (a): Distribution of $\beta_t a_t$, (b): Scatter plot for $\|\Delta_t v\|, \|\Delta_t p\|, \|r_t\|$

Building on the one-step contraction result in Lemma 3.1, we further establish a global stability tracking for Q-Delta, showing that the mixed readout error shows geometric decay and remains uniformly bounded over time, with the bound proportional to the magnitude of residual drifts consisting of target drift and prediction drift.

**Theorem 3.2** (Global stability and geometric tracking of Q-Delta). *Suppose the single-step contraction condition of Lemma 3.1 holds with constant $\rho \in (0, 1)$, i.e.,*

$$\|v_t - S_t x_t\| \le \rho \|v_t - S_{t-1}x_t\| \qquad \text{a.s. for all } t \ge 1.$$

*Define the pre-update and post-update prediction errors*

$$\tilde{e}_t := v_t - S_{t-1}x_t, \qquad e_t := v_t - S_t x_t.$$

*Define the residual drift*

$$r_t := \Delta_t v - \Delta_t p,$$

*where the $\Delta_t v := v_t - v_{t-1}$ is target drift and $\Delta_t p := S_{t-1}(x_t - x_{t-1})$ is prediction drift. Assume $r_t$ is uniformly bounded, then there exists $r < \infty$ such that*

$$\|r_t\| \le r \qquad \text{for all } t \ge 1.$$

*Then, almost surely for all $t \ge 1$,*

$$\|e_t\| \le \rho \|e_{t-1}\| + \rho r \le \rho^t \|e_0\| + \frac{1 - \rho^t}{1 - \rho} \rho r.$$

Theorem 3.2 establishes that Q-Delta induces a stable global tracking dynamics on mixed key–query prediction errors. As long as the one-step contraction condition $\beta_t a_t \in (0, 2)$ holds, the mixed readout error decays geometrically up to a bounded radius whose size is controlled by the magnitude

of the residual drift $r_t$, which aggregates both target drift $\Delta_t v$ and prediction drift $\Delta_t p$. Intuitively, $\Delta_t p := S_{t-1} \Delta x_t$ captures how changes in the mixed input $(k_t, q_t)$ induce variation in the model's joint prediction through the accumulated memory state.

Figure 3-(b) visualizes the relationship between these two drift terms $\Delta_t v$ and $\Delta_t p$, showing that prediction drift is typically smaller in magnitude than the corresponding target drift and remains concentrated near zero. This empirical behavior indicates that residual drift is largely dominated by target variation rather than prediction instability driven by readout key drift. Figure 3-(b) also implies that the residual drift terms in steady-state bound given in Theorem 3.2 are well-controlled in magnitude within range (0, 1.2), with its mean norm 0.538, yielding a tight and practically useful characterization of Q-Delta's error contraction dynamics.

Taken together, Q-Delta behaves as a stable online learner, ensuring transient error contraction and long-horizon stability under empirically verified conditions. This provides theoretical support for incorporating query-conditioned feedback into state evolution, and justifies its use as a principled state evolution mechanism beyond pure key–value association. Proofs for Lemma 3.1 and Theorem 3.2 are in Appendix C.2.

## 4. Experiments

### 4.1. Experimental Setup

All models are implemented based on pretraining framework flash-linear-attention (Yang & Zhang, 2024). We consider two model scales, 340M and 1.3B where the 340M models are pretrained on 15B tokens from the FineWeb-Edu (Penedo et al., 2024), while the 1.3B models are pretrained on 30B tokens. Training is performed on 4 NVIDIA RTX Pro 6000 (Blackwell) GPUs using mixed-precision arithmetic with bfloat16. We compare Q-Delta against RetNet (Sun et al., 2023), Mamba (Gu & Dao, 2024), Mamba2 (Dao & Gu, 2024), DeltaNet (Yang et al., 2025b), and GatedDeltaNet (Yang et al., 2025a). All baselines are reproduced on the same framework and trained under matched optimization settings to ensure fair comparison. We use the AdamW optimizer with cosine learning rate scheduling and gradient clipping, with a peak learning rate of $1 \times 10^{-3}$ for 340M models and $4 \times 10^{-4}$ for 1.3B models.

Figure 4-(a) shows the training loss curves of 340M-parameter models pretrained on 15B tokens. Q-Delta exhibits stable optimization behavior throughout training and achieves comparable or lower training loss relative to prior linear attention and state-space baselines. The zoomed region within box highlights the early training phase, where Q-Delta follows comparable or even faster convergence trajectory without introducing optimization instability.

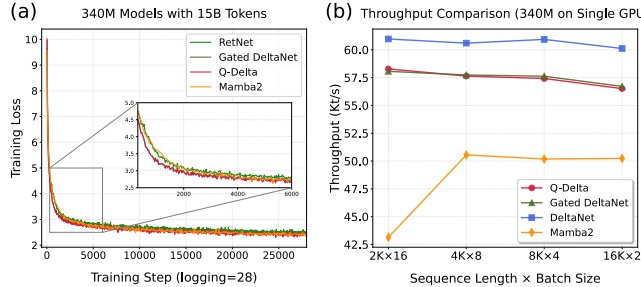

*Figure 4.* Training results on 340M Models. (a): Train loss curves over 28,600 steps (logging interval 28), with the box highlighting early-phase. (b): Single-GPU training throughput comparison across varying sequence length × batch size configurations.

Figure 4-(b) reports a single-GPU throughput comparison on 340M-parameter models. Throughput is measured as tokens processed per second by running 50 training steps on a single GPU, while varying the sequence length and batch size such that the total token count per step remains constant. We evaluate configurations from $(2048, 16)$ to $(16384, 2)$, matching increased sequence lengths with proportionally reduced batch sizes. Q-Delta achieves consistently high throughput across all configurations, closely matching delta-based baselines. In contrast, Mamba2 exhibits noticeably lower throughput, particularly at shorter sequence lengths. Overall, the results indicate that Q-Delta preserves the computational efficiency of delta-rule architectures while scaling robustly to longer sequence under practical training settings.

### 4.2. Evaluation

**Language Modeling.** We evaluate commonsense reasoning performance using LM Evaluation Harness (Sutawika et al., 2024) to test zero-shot language modeling capacity. Following standard practice, we report language modeling perplexity on LAMBADA (Paperno et al., 2016) and Wikitext (Merity et al., 2016), and zero-shot accuracy on multiple-choice reasoning benchmarks, including BoolQ (Clark et al., 2019), HellaSwag (Zellers et al., 2019), PIQA (Bisk et al., 2019), Arc-Easy, Arc-Challenge (Clark et al., 2018), WinoGrande (Sakaguchi et al., 2019), and OpenBookQA (Mihaylov et al., 2018).

From Table 2, across both model scales, Q-Delta consistently achieves strong zero-shot performance on commonsense reasoning benchmarks. At the 340M scale, Q-Delta attains the best average accuracy and improves over other baselines on various reasoning tasks, while maintaining competitive perplexity on both WikiText and LAMBADA. At 1.3B scale, Q-Delta further strengthens this trend, achieving the highest average score and leading performance on several benchmarks, notably on language modeling tasks, ARC-Challenge, HellaSwag, and BoolQ. These results indicate that incorporating query-conditioned feedback into state evolution improves both language modeling and zero-shot reasoning ability without task-specific adaptation.

*Table 2.* Zero-shot performance comparison of 340M and 1.3B models trained on FineWeb-Edu (Penedo et al., 2024). The commonsense Reasoning task is evaluated by lm-evaluation-harness (Gao et al., 2024). All reproduced by us. Best in **bold** and second-best underlined.

| Model | Lamb ppl. ↓ | Wiki ppl. ↓ | ARC$_E$ ↑ | ARC$_C$ ↑ | Hella. ↑ | Lamb. ↑ | PIQA ↑ | Wino. ↑ | BoolQ ↑ | OpenBook ↑ | Avg. ↑ |
|---|---|---|---|---|---|---|---|---|---|---|---|
| *340M parameters, 15B training tokens* | | | | | | | | | | | |
| RetNet | 52.29 | 31.36 | 57.07 | 28.41 | 38.71 | 27.36 | 66.54 | 49.41 | 56.88 | 32.00 | 44.55 |
| Mamba | 31.20 | 27.50 | 59.68 | **29.18** | **42.98** | 32.97 | 67.52 | 51.78 | 55.81 | 33.20 | 46.64 |
| Mamba2 | **30.35** | **26.60** | 59.30 | 29.01 | 42.13 | 33.67 | **68.01** | 52.33 | 51.90 | 33.80 | 46.27 |
| DeltaNet | 63.04 | 28.78 | 54.88 | 27.47 | 38.65 | 27.11 | 63.60 | 49.96 | 59.46 | 29.40 | 43.82 |
| Gated DeltaNet | 36.27 | 27.82 | **60.02** | 25.94 | 40.25 | 31.52 | 67.30 | 51.54 | 57.13 | **34.40** | 46.01 |
| **Q-Delta** | 32.67 | 26.89 | 59.51 | 28.50 | 41.61 | **33.90** | 67.63 | 52.88 | 59.48 | **34.40** | **47.24** |
| *1.3B parameters, 30B training tokens* | | | | | | | | | | | |
| RetNet | 21.84 | 22.45 | 63.68 | 33.36 | 47.73 | 38.70 | 69.04 | 52.72 | 60.61 | 36.60 | 50.31 |
| Mamba | 16.98 | 19.89 | 68.10 | 36.18 | 53.44 | 40.77 | **72.20** | 55.01 | 55.63 | 37.80 | 52.39 |
| Mamba2 | 17.40 | 19.47 | **69.87** | 36.35 | 53.24 | 40.68 | 70.29 | **56.04** | 55.81 | 37.40 | 52.46 |
| DeltaNet | 16.64 | 19.77 | 67.63 | 34.47 | 51.09 | 41.78 | 70.95 | 54.70 | 61.19 | 38.40 | 52.53 |
| Gated DeltaNet | 15.32 | 19.61 | 68.60 | 33.28 | 52.60 | **43.80** | 70.84 | 54.78 | 59.42 | **38.80** | 52.77 |
| **Q-Delta** | **15.19** | **19.21** | 68.27 | **36.60** | **53.46** | 43.28 | 71.44 | 54.93 | **61.41** | 38.40 | **53.47** |

*Table 3.* Retrieval performance on the synthetic S-NIAH benchmark from RULER Hsieh et al. (2024), evaluated on 1.3B models. Results are reported under varying context lengths (1K, 2K, and 4K tokens). Best results are shown in **bold** and second-best in underlined.

| Model | S-NIAH-1 (pass-key retrieval) | | | S-NIAH-2 (number in haystack) | | | S-NIAH-3 (uuid in haystack) | | | Avg. |
|---|---|---|---|---|---|---|---|---|---|---|
| | 1K | 2K | 4K | 1K | 2K | 4K | 1K | 2K | 4K | |
| RetNet | 96.6 | 27.8 | 7.4 | 99.4 | 60.8 | 24.4 | 20.0 | 5.2 | 1.2 | 38.09 |
| Mamba | 99.8 | 99.6 | 87.0 | 98.8 | 92.8 | 50.8 | 22.0 | 12.0 | 0.8 | 62.62 |
| Mamba2 | **100.0** | 99.8 | 99.0 | 99.8 | 95.4 | 57.0 | 76.0 | 50.6 | 11.6 | 76.58 |
| DeltaNet | **100.0** | **100.0** | **100.0** | 99.8 | 93.6 | 49.6 | 87.4 | **75.8** | 25.4 | 81.29 |
| Gated DeltaNet | **100.0** | **100.0** | **100.0** | 100.0 | **99.8** | 76.6 | 83.8 | 70.0 | 21.4 | 83.51 |
| **Q-Delta** | **100.0** | **100.0** | **100.0** | 100.0 | 99.4 | **94.2** | **94.6** | 74.0 | **48.0** | **90.02** |

*Table 4.* Retrieval performance on real-world recall-intensive tasks from Arora et al. (2024), evaluated with 340M-parameter models. All inputs are truncated to a context length of 2K tokens and formatted in a cloze-style next-token prediction setting. Best results are shown in **bold** and second-best in underlined.

| Models | SWDE | SQD | FDA | TQA | NQ | Drop |
|---|---|---|---|---|---|---|
| Mamba | 17.1 | 43.6 | 6.4 | **55.2** | 17.5 | 26.4 |
| Mamba2 | 29.1 | 55.3 | 18.4 | 49.1 | 18.2 | 33.4 |
| DeltaNet | 22.9 | 51.5 | 16.7 | 45.0 | 15.2 | 28.2 |
| Gated DeltaNet | 29.1 | **56.0** | 18.9 | 48.9 | 18.2 | **34.2** |
| **Q-Delta** | **31.6** | 52.3 | **22.0** | 48.6 | **18.4** | 33.2 |

**Real and synthetic retrieval.** We evaluate retrieval capabilities using both real-world and synthetic benchmarks. For real-world retrieval, we adopt the recall-intensive tasks (Arora et al., 2024) and evaluate on 340M models. All real-world retrieval inputs are truncated to a maximum context length of 2K tokens. For synthetic retrieval, we evaluate 1.3B-parameter models on the S-NIAH (Synthetic Needle-In-A-Haystack) benchmark (Hsieh et al., 2024), which measures model's ability to retrieve sparse target information embedded at varying positions within long contexts. We report results under context lengths of 1K, 2K, and 4K tokens to assess generalization beyond the training context. Together, these benchmarks evaluate complementary aspects of retrieval, ranging from structured real-world recall to

controlled long-context information extraction.

On real-world recall-intensive tasks (Table 4), Q-Delta consistently matches or outperforms prior linear RNN models, achieving the best average score across tasks. These results suggest that incorporating query-conditioned feedback into state evolution improves both controlled synthetic retrieval and practical real-world recall, while maintaining linear-time scalability. On the synthetic S-NIAH benchmark (Table 3), Q-Delta achieves the highest average accuracy among all linear recurrent baselines, with near-perfect performance on pass-key retrieval (S-NIAH-1) across all evaluated context lengths. Notably, Q-Delta substantially improves performance on the more challenging number-in-haystack and UUID-in-haystack tasks (S-NIAH-2 and S-NIAH-3), particularly at longer contexts up to 4K tokens, indicating stronger robustness to sparse information retrieval especially as context length increases.

**Ablation Studies.** Given Q-Delta recurrence rule $S_t = \alpha_t S_{t-1}\left(I - \beta_t(k_t + \lambda_t q_t)k_t^\top\right) + \beta_t v_t k_t^\top$, the query-conditioned correction is governed solely by $\lambda_t$, which scales the query-feedback term $\lambda_t q_t k_t^\top$. Since the query-based state correction is our central contribution, $\lambda_t$ is the natural ablation target, and we additionally ablate the decay factor $\alpha_t$ (Table 5). Across fixed scalar values, performance

*Table 5.* Ablation on the query-feedback coefficient $\lambda$ for 340M Q-Delta. Scalar $\lambda$ tests fixed query-feedback strength versus adaptive learnable modulation, No state decay removes the recurrent forget/decay gate ($\alpha_t = 1$), and No gating uses full query correction ($\lambda_t = 1$), disabling adaptive gating of the query-feedback term.

| $\lambda$ | Wiki ppl. $\downarrow$ | Lamb ppl. $\downarrow$ | Avg Acc. (8 tasks) $\uparrow$ |
|---|---|---|---|
| Learnable $\lambda_t$ (Q-Delta) | 26.89 | 32.67 | 47.24 |
| Scalar $\lambda = 0.2$ | 26.96 | 35.39 | 46.99 |
| Scalar $\lambda = 0.5$ | 26.86 | 33.31 | 47.20 |
| Scalar $\lambda = 0.8$ | 26.61 | 33.58 | 46.42 |
| No state decay ($\alpha_t = 1.0$) | 26.52 | 32.97 | 45.86 |
| No gating ($\lambda_t = 1.0$) | 26.55 | 35.21 | 46.36 |

is relatively robust, with $\lambda = 0.5$ giving the best scalar result (47.20 average accuracy) while learning $\lambda_t$ end-to-end yields the best overall performance, maintaining strong perplexity. This indicates that adaptively modulating the query-feedback strength is beneficial.

Removing the decay factor ($\alpha_t = 1$) lowers accuracy to 45.86 but remains clearly above its most direct non-decay baseline, DeltaNet (43.82 at the same 340M scale), indicating that query feedback is beneficial independent of the gating mechanism. Overall, query feedback contributes consistently across settings, and allowing the model to tune its strength gives the best trade-off.

## 5. Conclusion

This work reconsiders a core assumption in linear attention and recurrent sequence models: that state evolution is governed solely by key–value association, with queries confined to passive readout. We observe that the query is the direction along which the state is read out, so the query-conditioned value prediction $\hat{o}_t = S_{t-1}q_t$ is a readout-aligned state correction signal that conventional delta-rule updates leave uncorrected. Building on this, we propose Q-Delta, a query-aware delta rule that injects this prediction error into state evolution while preserving linear-time efficiency, and we establish theoretical justification that the resulting mixed key–query dynamics are stable under mild, empirically verified conditions. Q-Delta consistently improves over strong linear-attention and SSM baselines, showing that incorporating query into recurrent state update is an effective way to move beyond pure key–value association.

## Acknowledgements

This work was partly supported by the Institute for Information & Communications Technology Planning & Evaluation (IITP) grants funded by the Korean government (MSIT) (No. RS-2026-25526850, High-Efficiency Neural Networks for Artificial General Intelligence, 33%; No.2022-0-00857, Development of Financial and Economic Digital Twin Platform based on AI and Data, 33%; No. RS-2025-25442149, LG AI STAR Talent Development Program for Leading Large-Scale Generative AI Models in the Physical AI Domain, 1%), and Samsung Research Funding & Incubation Center of Samsung Electronics under Project Number SRFC-IT2402-08, 33%.

## Impact Statement

This work proposes Q-Delta, a novel query-aware delta rule that enriches linear-time sequential models, enabling rich state dynamics by integrating complementary key–query signals into state evolution. The research supports scalable language modeling and long-context applications, improving expressivity and interpretability of linear attention frameworks. As with other large-scale sequence models, potential risks such as misuse for misinformation generation or unintended memorization of sensitive data may arise in wide applications, so maintaining responsible training, evaluation, and deployment practices is important.

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

## A. Datasets

**Commonsense Reasoning.** We evaluate on zero-shot commonsense reasoning benchmarks. For multiple-choice tasks, we report task accuracy on PIQA (Bisk et al., 2019), HellaSwag (Zellers et al., 2019), WinoGrande (Sakaguchi et al., 2019), ARC-Easy, ARC-Challenge (Clark et al., 2018), OpenBookQA (Mihaylov et al., 2018), and BoolQ (Clark et al., 2019), as well as language modeling tasks on WikiText (Merity et al., 2016) and LAMBADA (Paperno et al., 2016). All evaluations are conducted in a zero-shot setting using the LM Evaluation Harness (Gao et al., 2024).

**In-Context Retrieval.** To assess retrieval capacities, we consider both synthetic and real-world in-context retrieval benchmarks. For synthetic evaluation, we use the Needle-In-A-Haystack Single (NIAH-S) benchmark from RULER (Hsieh et al., 2024), which consists of three tasks, passkey retrieval (S-NIAH-1), numerical needle retrieval (S-NIAH-2), and word-based needle retrieval (S-NIAH-3). These tasks evaluate a model's ability to recover sparse target information embedded at arbitrary positions within long contexts. For real-world retrieval, we follow the evaluation protocol introduced by (Arora et al., 2024). These include SWDE (Lockard et al., 2019) for structured HTML relation extraction, FDA (Arora et al., 2023) for key–value retrieval from PDFs, and several question-answering datasets such as SQuAD (Rajpurkar et al., 2018), TriviaQA (Joshi et al., 2017), DROP (Dua et al., 2019), and Natural Questions (NQ) (Kwiatkowski et al., 2019). All real-world retrieval inputs are truncated to a maximum context length of 2K tokens. Since our pretrained models are not instruction-tuned, we adopt cloze completion prompts as provided by prior work (Yang et al., 2025a).

## B. Query-Feedback Coefficient $\lambda_t$

The coefficient $\lambda_t$ is computed per head from the hidden state as $\lambda_t = \sigma(W_\lambda h_t + b)$, where $\sigma$ is the logistic sigmoid, $W_\lambda \in \mathbb{R}^{H \times d_{\text{model}}}$ with $H$ the number of heads, and $b$ is a scalar bias initialized to $-0.8$. This adds the single projection $W_\lambda$ over the gated delta rule, introducing no additional recurrent state or $q, k, v$ transformation.

## C. Theoretical Derivations

### C.1. Query for Value Prediction

We consider the linear recurrence

$$S_t = S_{t-1}P_t + \eta_t v_t k_t^\top, \qquad S_0 = 0, \tag{28}$$

where $S_t \in \mathbb{R}^{d_v \times d_k}$, $P_t \in \mathbb{R}^{d_k \times d_k}$ is a linear operator, $v_t \in \mathbb{R}^{d_v}$, $k_t \in \mathbb{R}^{d_k}$, and $\eta_t \in \mathbb{R}$.

We show by induction that for all $t \geq 1$, there exist vectors $\{b_{\tau,t}\}_{\tau \leq t} \subset \mathbb{R}^{d_k}$ such that

$$S_t = \sum_{\tau=1}^{t} v_\tau b_{\tau,t}^\top. \tag{29}$$

For $t = 0$, $S_0 = 0$ and the claim holds trivially. Assume that Eq. (29) holds for $S_{t-1}$, i.e.,

$$S_{t-1} = \sum_{\tau=1}^{t-1} v_\tau b_{\tau,t-1}^\top.$$

Substituting into Eq. (28) gives

$$\begin{aligned}
S_t &= S_{t-1}P_t + \eta_t v_t k_t^\top \\
&= \left( \sum_{\tau=1}^{t-1} v_\tau b_{\tau,t-1}^\top \right) P_t + \eta_t v_t k_t^\top \\
&= \sum_{\tau=1}^{t-1} v_\tau \left( b_{\tau,t-1}^\top P_t \right) + v_t (\eta_t k_t)^\top.
\end{aligned}$$

Using the identity $b^\top P = (P^\top b)^\top$, define

$$b_{\tau,t} := P_t^\top b_{\tau,t-1} \in \mathbb{R}^{d_k}, \quad \tau = 1, \ldots, t-1, \qquad b_{t,t} := \eta_t k_t.$$

Then

$$S_t = \sum_{\tau=1}^{t} v_\tau b_{\tau,t}^\top,$$

which completes the induction.

For any query $q_t \in \mathbb{R}^{d_k}$, the query-conditioned prediction satisfies

$$\hat{o}_t := S_{t-1} q_t = \sum_{\tau=1}^{t-1} v_\tau (b_{\tau,t-1}^\top q_t)$$

$$= \sum_{\tau=1}^{t-1} \gamma_{\tau,t} v_\tau, \qquad \gamma_{\tau,t} := b_{\tau,t-1}^\top q_t \in \mathbb{R}.$$

## C.2. Stability Analysis of Q-Delta Dynamics

**Lemma 3.1** (One-step contraction of mixed prediction error under Q-Delta). *Let $k_t, q_t \in \mathbb{R}^d$ and $\lambda_t \in [0, 1]$, and define the mixed input $x_t := k_t + \lambda_t q_t$. Consider the Q-Delta update*

$$S_t = S_{t-1} + \beta_t (v_t - S_{t-1} x_t) k_t^\top, \qquad \beta_t \in (0, 1].$$

*Assume that the scalar alignment $a_t := k_t^\top x_t$ satisfies $\beta_t a_t \in (0, 2)$ almost surely, and define*

$$\rho := \sup_t |1 - \beta_t a_t| \in (0, 1).$$

*Then the mixed prediction error contracts in one step:*

$$\|v_t - S_t x_t\| \leq \rho \|v_t - S_{t-1} x_t\| \quad \text{almost surely for all } t.$$

*Proof.* Let $x_t := k_t + \lambda_t q_t$ and consider the Q-Delta update

$$S_t = S_{t-1} + \beta_t (v_t - S_{t-1} x_t) k_t^\top.$$

Right-multiply both sides by $x_t$ to obtain

$$S_t x_t = S_{t-1} x_t + \beta_t (v_t - S_{t-1} x_t) k_t^\top x_t.$$

Define the scalar alignment $a_t := k_t^\top x_t$. Then

$$S_t x_t = S_{t-1} x_t + \beta_t a_t (v_t - S_{t-1} x_t).$$

Rearranging gives the exact identity

$$v_t - S_t x_t = v_t - S_{t-1} x_t - \beta_t a_t (v_t - S_{t-1} x_t) = (1 - \beta_t a_t)(v_t - S_{t-1} x_t).$$

Taking norms yields

$$\|v_t - S_t x_t\| = |1 - \beta_t a_t| \, \|v_t - S_{t-1} x_t\|.$$

By assumption $\beta_t a_t \in (0, 2)$ almost surely, hence $|1 - \beta_t a_t| < 1$. With $\rho := \sup_t |1 - \beta_t a_t| \in (0, 1)$, we therefore have

$$\|v_t - S_t x_t\| \leq \rho \|v_t - S_{t-1} x_t\| \qquad \text{almost surely for all } t.$$

$\square$

**Theorem 3.2** (Global stability and geometric tracking of Q-Delta). *Suppose the single-step contraction condition of Lemma 3.1 holds with constant $\rho \in (0, 1)$, i.e.,*

$$\|v_t - S_t x_t\| \leq \rho \|v_t - S_{t-1} x_t\| \qquad \text{a.s. for all } t \geq 1.$$

*Define the pre-update and post-update prediction errors*

$$\tilde{e}_t := v_t - S_{t-1}x_t, \qquad e_t := v_t - S_t x_t.$$

*Define the residual drift*

$$r_t := \Delta_t v - \Delta_t p,$$

*where the $\Delta_t v := v_t - v_{t-1}$ is target drift and $\Delta_t p := S_{t-1}(x_t - x_{t-1})$ is prediction drift. Assume $r_t$ is uniformly bounded, then there exists $r < \infty$ such that*

$$\|r_t\| \leq r \qquad for\ all\ t \geq 1.$$

*Then, almost surely for all $t \geq 1$,*

$$\|e_t\| \leq \rho\|e_{t-1}\| + \rho r \ \leq\ \rho^t\|e_0\| + \frac{1-\rho^t}{1-\rho}\,\rho r.$$

*Proof.* By definition of the Q-Delta update,

$$S_t = S_{t-1} + \beta_t\big(v_t - S_{t-1}x_t\big)k_t^\top.$$

Right-multiplying by $x_t$ gives

$$S_t x_t = S_{t-1}x_t + \beta_t\big(v_t - S_{t-1}x_t\big)k_t^\top x_t = S_{t-1}x_t + \beta_t a_t\big(v_t - S_{t-1}x_t\big),$$

where $a_t := k_t^\top x_t$. Rearranging and using the definitions

$$\tilde{e}_t := v_t - S_{t-1}x_t, \qquad e_t := v_t - S_t x_t,$$

yields the exact identity

$$e_t = v_t - S_t x_t = v_t - \Big(S_{t-1}x_t + \beta_t a_t(v_t - S_{t-1}x_t)\Big) = (1 - \beta_t a_t)\tilde{e}_t.$$

Taking norms and using $\rho := \sup_t |1 - \beta_t a_t| < 1$ gives

$$\|e_t\| \leq \rho\,\|\tilde{e}_t\| \qquad \forall t. \tag{30}$$

Next, starting from $\tilde{e}_t = v_t - S_{t-1}x_t$, add and subtract $v_{t-1}$ and $S_{t-1}x_{t-1}$:

$$\tilde{e}_t = (v_{t-1} - S_{t-1}x_{t-1}) + (v_t - v_{t-1}) - S_{t-1}(x_t - x_{t-1}).$$

Recognize that $v_{t-1} - S_{t-1}x_{t-1} = e_{t-1}$, and define the target drift $\Delta_t v := v_t - v_{t-1}$ and the prediction drift $\Delta_t p := S_{t-1}(x_t - x_{t-1})$ to obtain

$$\tilde{e}_t = e_{t-1} + \Delta_t v - \Delta_t p = e_{t-1} + r_t,$$

where the residual drift is $r_t := \Delta_t v - \Delta_t p$. Taking norms and applying the triangle inequality gives

$$\|\tilde{e}_t\| \leq \|e_{t-1}\| + \|r_t\|. \tag{31}$$

Combining (30) and (31) yields

$$\|e_t\| \leq \rho\big(\|e_{t-1}\| + \|r_t\|\big).$$

Under the uniform boundedness assumption, i.e., $\|r_t\| \leq r$ for all $t \geq 1$, we obtain the recurrence

$$\|e_t\| \leq \rho\|e_{t-1}\| + \rho r.$$

Unrolling this gives

$$\begin{aligned}
\|e_t\| &\leq \rho\|e_{t-1}\| + \rho r \\
&\leq \rho\big(\rho\|e_{t-2}\| + \rho r\big) + \rho r \\
&= \rho^2\|e_{t-2}\| + \rho r(\rho + 1) \\
&\vdots \\
&\leq \rho^t\|e_0\| + \rho r \sum_{j=0}^{t-1}\rho^j.
\end{aligned}$$

Solving the geometric series further gives

$$\|e_t\| \le \rho^t \|e_0\| + \frac{1-\rho^t}{1-\rho}\, \rho r.$$

$\square$

## D. Chunkwise Parallelization of Q-Delta

### D.1. WY representation

Here we drive $P^r$ and $G^r$ in terms of $\gamma$, $w^i$, and $u^i$. For clarity, we drop the chunk index $[t]$ and write $k_i := k_{t_i}$, $x_i := x_{t_i}$, $v_i := v_{t_i}$, $\alpha_i := \alpha_{t_i}$, $\beta_i := \beta_{t_i}$, $w^i := w^i_{[t]}$, $u^i := u^i_{[t]}$. Define

$$P_i := I - \beta_i x_i k_i^\top, \qquad \gamma^r := \prod_{j=1}^r \alpha_j, \quad (\gamma^0 := 1).$$

Recall that $F^r = \prod_{i=1}^r \alpha_i P_i$ in Eq. (21), which gives

$$F^r = \Big(\prod_{i=1}^r \alpha_i\Big)\Big(\prod_{i=1}^r P_i\Big) = \gamma^r P^r, \qquad P^r := \prod_{i=1}^r P_i.$$

Assume inductively that

$$P^{r-1} = I - \sum_{i=1}^{r-1} w^i k_i^\top.$$

Multiplying by $P_r$ gives

$$
\begin{aligned}
P^r &= P^{r-1} P_r \\
&= \Big(I - \sum_{i=1}^{r-1} w^i k_i^\top\Big)\Big(I - \beta_r x_r k_r^\top\Big) \\
&= I - \sum_{i=1}^{r-1} w^i k_i^\top - \beta_r x_r k_r^\top + \beta_r \sum_{i=1}^{r-1} w^i (k_i^\top x_r) k_r^\top \\
&= I - \sum_{i=1}^{r-1} w^i k_i^\top - \beta_r \Big(x_r - \sum_{i=1}^{r-1} w^i (k_i^\top x_r)\Big) k_r^\top.
\end{aligned}
$$
(32)

Define

$$w^r := \beta_r \Big(x_r - \sum_{i=1}^{r-1} w^i (k_i^\top x_r)\Big),$$

which then yields

$$P^r = I - \sum_{i=1}^r w^i k_i^\top.$$

Recall that the additive term in the chunkwise expansion (Eq. (21)) is

$$G^r = \sum_{i=1}^r \beta_i v_i k_i^\top \prod_{j=i+1}^r \alpha_j P_j.$$

Splitting the product,

$$\prod_{j=i+1}^r \alpha_j P_j = \Big(\prod_{j=i+1}^r \alpha_j\Big)\Big(\prod_{j=i+1}^r P_j\Big) = \frac{\gamma^r}{\gamma^i} \prod_{j=i+1}^r P_j,$$

and therefore

$$G^r = \sum_{i=1}^{r} \frac{\gamma^r}{\gamma^i} \beta_i v_i k_i^\top \prod_{j=i+1}^{r} P_j.$$

From above, $G^r$ satisfies the recursion

$$G^r = \alpha_r G^{r-1} P_r + \beta_r v_r k_r^\top, \qquad G^0 = 0.$$

Assume inductively that

$$G^{r-1} = \sum_{i=1}^{r-1} \frac{\gamma^{r-1}}{\gamma^i} \tilde{u}^i k_i^\top.$$

Multiplying by $\alpha_r$ gives

$$\alpha_r G^{r-1} = \sum_{i=1}^{r-1} \frac{\gamma^r}{\gamma^i} \tilde{u}^i k_i^\top,$$

since $\gamma^r = \alpha_r \gamma^{r-1}$. Multiplying by $P_r$ and adding the new term yields

$$G^r = \Big( \sum_{i=1}^{r-1} \frac{\gamma^r}{\gamma^i} \tilde{u}^i k_i^\top \Big) \Big( I - \beta_r x_r k_r^\top \Big) + \beta_r v_r k_r^\top$$

$$= \sum_{i=1}^{r-1} \frac{\gamma^r}{\gamma^i} \tilde{u}^i k_i^\top - \beta_r \sum_{i=1}^{r-1} \frac{\gamma^r}{\gamma^i} \tilde{u}^i (k_i^\top x_r) k_r^\top + \beta_r v_r k_r^\top$$

$$= \sum_{i=1}^{r-1} \frac{\gamma^r}{\gamma^i} \tilde{u}^i k_i^\top + \beta_r \Big( v_r - \sum_{i=1}^{r-1} \frac{\gamma^r}{\gamma^i} \tilde{u}^i (k_i^\top x_r) \Big) k_r^\top. \tag{33}$$

To match the desired form $G^r = \sum_{i=1}^{r} \frac{\gamma^r}{\gamma^i} \tilde{u}^i k_i^\top$, we define

$$u^r := \beta_r \Big( v_r - \sum_{i=1}^{r-1} \frac{\gamma^r}{\gamma^i} \tilde{u}^i (k_i^\top x_r) \Big).$$

Therefore, we have following relations,

$$P^r = I - \sum_{i=1}^{r} w^i k_i^\top, \qquad w^r = \beta_r \Big( x_r - \sum_{i=1}^{r-1} w^i (k_i^\top x_r) \Big),$$

$$G^r = \sum_{i=1}^{r} \frac{\gamma^r}{\gamma^i} \tilde{u}^i k_i^\top, \qquad u^r = \beta_r \Big( v_r - \sum_{i=1}^{r-1} \frac{\gamma^r}{\gamma^i} \tilde{u}^i (k_i^\top x_r) \Big),$$

which completes the extended WY representation.

## D.2. UT transform

The extended WY recursion defining $\tilde{u}^r$ is

$$\tilde{u}^r = \beta_r \left( v_r - \sum_{i=1}^{r-1} \frac{\gamma^r}{\gamma^i} \tilde{u}^i (k_i^\top x_r) \right). \tag{34}$$

We now rewrite this in matrix form. Let

$$\widetilde{U} \in \mathbb{R}^{C \times d_v}, \quad V \in \mathbb{R}^{C \times d_v}, \quad K \in \mathbb{R}^{C \times d_k}, \quad X \in \mathbb{R}^{C \times d_k}$$

stack rows $\tilde{u}^r, v_r, k_r, x_r$ respectively. Let $B := \mathrm{diag}(\beta) \in \mathbb{R}^{C \times C}$. Define $\Gamma \in \mathbb{R}^{C \times C}$ by

$$\Gamma_{ri} := \begin{cases} \gamma^r / \gamma^i, & r > i, \\ 0, & r \le i, \end{cases} \qquad \text{(strictly lower triangular)}.$$

Now define

$$L_\gamma := \text{strictLower}\big(B(\Gamma \odot KX^\top)\big) \in \mathbb{R}^{C \times C},$$

equivalently, for each row $r$,

$$(L_\gamma \widetilde{U})_{r,:} = \sum_{i<r} \beta_r \Gamma_{ri}(k_i^\top x_r)\tilde{u}^{i\top},$$

Then we can rewrite Eq. (34) for all $r$ as

$$\widetilde{U} + L_\gamma \widetilde{U} = BV,$$

hence

$$\widetilde{U} = (I + L_\gamma)^{-1}BV. \tag{35}$$

Therefore, defining the UT transform matrix

$$T_\gamma := (I + L_\gamma)^{-1}B,$$

we obtain the matrix form

$$\widetilde{U} = T_\gamma V.$$

