# OpenReview forum: "Q-Delta: Beyond Key–Value Associative State Evolution"
_ICML.cc/2026/Conference — ICML 2026 regular_

### Official Review · Reviewer_fzVL · 2026-02-17

**Soundness:** 3
**Presentation:** 2
**Significance:** 3
**Originality:** 3
**Overall Recommendation:** 5
**Confidence:** 4

**Summary:**

The paper proposes Q-Delta, a modification of the delta-rule that uses the query readout as a prediction signal during the memory update, not only the key readout.
The update writes mainly what the current memory cannot already produce, reducing redundant writes and interference.

Intuitively, without subtracting the query-based prediction, each token tends to write a full value even when the memory could already reconstruct it, causing drift. Q-Delta avoids rewriting already represented content, making better use of limited memory capacity.

The paper provides a stability analysis, a chunkwise-parallel Triton implementation, and shows consistent gains over delta/linear-time baselines on language modeling, zero-shot benchmarks, and long-context retrieval (S-NIAH up to 4K).

**Compliance With Llm Reviewing Policy:**

Affirmed.

**Final Justification:**

The authors provided additional empirical support for stability, extended evaluation to 8K, and clarified why query-conditioned updates do not introduce harmful interference (in practice).
Overall, I continue to find the paper technically solid, well motivated, and empirically convincing. I therefore keep my score at 5 (Accept).

**Key Questions For Authors:**

1. Do the retrieval gains persist on harder and longer-context evaluations than S-NIAH up to 4K, especially at the 1.3B scale (e.g., where S-NIAH-1 at 4K shows little/no gap vs GDN)?

2. How sensitive is Q-Delta to noisy or misaligned queries—can query-conditioned updates increase interference, and is this failure mode stress-tested beyond the lambda robustness ablations?

**Limitations:**

The paper lacks a dedicated limitations paragraph, although limitations are implicitly discussed, such as the update’s non-GD nature.

**Strengths And Weaknesses:**

Strengths

1. **Clear motivation:** frames the query readout (state times query) as a structured value prediction (a mixed aggregation of past values) rather than a passive readout, motivating query-conditioned state evolution.
2. **Principled, minimal change:** adds query-conditioned feedback via a mixed error update that combines a key-based prediction and a query-based prediction.
3. **Evidence the extra signal is non-redundant:** cosine/PCA analysis suggests the query-based prediction captures value components not well represented by the key-based prediction, supporting mixed-error updates rather than "more of the same".
4. **Theory for a non-GD update:** explicitly note the update is not strict gradient descent, then provide stability results.
5. **Hardware-aware implementation:** derives a chunkwise-parallel formulation and provides Triton kernels. They show throughput close to delta-based baselines.


Weaknesses

1. **Stability relies on an empirical condition:** contraction depends on an observed alignment/step-size condition; the paper supports this empirically, but it remains an assumption about training dynamics rather than a first-principles guarantee.
2. **Context-length coverage is still modest:** retrieval generalization is shown up to 4K on S-NIAH; For 1.3M models I'd expect more challenging results. For example, S-NIAH-1 4K shows no difference with GDN.
3. **Extra coupling may be task-dependent:** feeding the query back into the memory update helps when the query is predictive of needed memory content; when queries are noisy/misaligned it could add interference, an edge case not deeply stress-tested beyond robustness ablations.

---

> ### Author Rebuttal · Authors · 2026-03-31
>
> We deeply appreciate reviewer fzVL for careful reading and raising the constructive and valuable feedbacks, which help us sharpen both the presentation and the technical positioning of our work.
>
> ### **W1. Empirical conditions on theoretical stability analysis**
>
> - While we admit that our stability result relies on an empirically verified condition, we want to emphasize that the empirical support is not loose in practice. The paper shows that $\beta_t a_t$ is tightly concentrated within the contraction regime $(0,2)$ throughout training, with mean $0.043$, while the residual drift that determines the bounded-error radius is also well controlled, with mean norm $0.538$. Thus, although the theorem is conditional, its underlying condition is directly measurable and consistently satisfied in practice.
>
> - To further substantiate this point, we additionally provide the empirical results that actually show the well-controlled tracking of value prediction error throughout entire training on 340M models.
>
>     - [Figure 6. Value prediction error trace throughout training on 340 models](https://anonymous.4open.science/r/ICML-2026-Discussion-2CD2/Additional_Figures.pdf)
>
>     - As shown in the figure, Q-Delta exhibits stable bounded convergence of value-prediction error across all three targets, $\|v_t-S_{t-1}q_t\|_2$, $\|v_t-S_{t-1}k_t\|_2$, and $\|v_t-S_{t-1}x_t\|_2$, with consistently lower error throughout training compared to Gated DeltaNet.
>
>     - We view this as complementary evidence to the theorem, beyond showing that the empirical condition is tightly satisfied, the new tracking plot shows that Q-Delta indeed operates in a practically stable regime over the full course of training, yielding better-controlled value-prediction dynamics than a key-only delta-rule update.
>
> ### **W2. Longer context generalization**
>
> - We thank the reviewer for pointing out the importance of evaluating in longer context regimes. To address this concern, we have conducted additional S-NIAH experiments at 8K context length, extending beyond the original 4K setting reported in Table 3.
>
>     - [Table 7. Retrieval performance on  S-NIAH benchmark on context length 8K](https://anonymous.4open.science/r/ICML-2026-Discussion-2CD2/Table%207.pdf)
>
> - The results show Q-Delta continues to demonstrate consistent and in fact stronger advantages in longer contexts, showing that the stable context evolution of Q-Delta persists under harsher condition.
>
> ### **W3. Concerns on noisy query**
>
> - We agree that noisy or misaligned queries can interfere the state evolution process. In practice, however, the query is not an external signal, rather it is produced from the same input representation as the corresponding key and value. Hence, query noise is less a query-specific issue than a general under-convergence issue of the representation itself, especially during the early phase of training.
>
> - Although we cannot fully prevent the noisy query inputs, we want to note that query feedback is only injected through the gated corrective target $\lambda_t q_t$. The fact that learnable $\lambda_t$ performs best as shown in ablation, while the ungated case $\lambda_=1.0$ is worse, suggests that the model already learns to use query feedback selectively and to suppress harmful interference when needed.
>
>     - For your information, we provide the actual distribution of $\lambda_t$ throughout pretraining on 340M Q-Delta.
>     - [Figure 5. Distribution of $\lambda_t$](https://anonymous.4open.science/r/ICML-2026-Discussion-2CD2/Additional_Figures.pdf)
>
>     - We observe that $\lambda_t$ is broadly distributed with a mean around 0.38, consistently taking nontrivial values across layers rather than collapsing toward zero. Having a broad, non-degenerate distribution, this implies that the model is actively using query feedback, but in a selective, input-dependent way.

---

> > ### Author Rebuttal · Reviewer_fzVL · 2026-04-01
> >
> > My concerns have been addressed. I thank the authors for the answers. I keep my score.

---

> > > ### Author Response · Authors · 2026-04-01
> > >
> > > We're glad that our response addressed your concerns. Thank you for carefully reading our rebuttal and your thoughtful acknowledgement.

---

### Official Review · Reviewer_gT2T · 2026-02-28

**Soundness:** 3
**Presentation:** 1
**Significance:** 1
**Originality:** 1
**Overall Recommendation:** 4
**Confidence:** 3

**Summary:**

The paper emphasizes that the query should participate in the state evolution of linear Transformers, rather than merely being used for the readout operation. Consequently, it introduces a query-related term into the state update. Experiments demonstrate that this approach yields marginal gains.

**Compliance With Llm Reviewing Policy:**

Affirmed.

**Final Justification:**

The rebuttal resolves my concerns, and I have raised my score correspondingly.

**Key Questions For Authors:**

1. Why is the term $\lambda_t\hat{o}_t$ added to Equation (18)?
2. What is the percentile for the accuracy of the experimental results?
3. What is the distribution of learnable $\lambda_t$’s values? Are they close to zero?

**Limitations:**

No. Limited model scale: While the design is generalizable, it is unclear how Q-Delta scales to larger models (e.g., 7B).

**Strengths And Weaknesses:**

## Strengths

- Soundness: The proofs are correct. Figure 3 verifies the conditions required for the lemma and the theorem.
- The linear Transformers are promising.

## Weaknesses

1. Unclear Motivation (Presentation): The motivation is confusing. Figure 2 shows that the similarity between $S_{t-1}q_t$ and $S_{t-1}k_t$ is rare, but this is a typical manifestation of attention sparsity: only a small fraction of $\langle q_t, k_t\rangle$ pairs have high values, while the majority have low values. (Question 1)
2. Marginal Improvement （Significance): The experimental results and train loss curves are close. (Question 2)
3. Useless Design (Originality): Table 5 shows that the smaller the $\lambda$, the greater the average accuracy. Perhaps $\lambda=0$ is optimal, but that would make Q-Delta identical to GatedDeltaNet. (Question 3)

Typos:
- Line 137, fovors -> favors
- Equations (19) and (20), $\lambda q_t k_t^\top$ -> $\lambda_t q_t k_t^\top$
- Figure 4(a), GPCK -> Q-Delta

---

> ### Author Rebuttal · Authors · 2026-03-31
>
> We sincerely thank the reviewer gT2T for careful reading and raising the constructive and valuable feedbacks, which help us sharpen both the presentation and the technical positioning of our work.
>
> ### **W1. Relation between Figure 2 and motivation**
>
> - We deeply appreciate the reviewer for raising this important point. We firstly want to clarify that Figure 2 was not meant to give the primary justification of Q-Delta, and we agree that the low similarity between $S_{t-1}q_t$ and $S_{t-1}k_t$ can reflect the attention sparsity.
>
> - With this clarification, our core motivation is that $q_t$ is not an arbitrary probe, but the actual readout direction by which the recurrent state is ultimately consumed. Given Eq. (16) that $S_{t-1}q_t$ should be viewed as the model's own memory-based value prediction along the very direction that determines the final output, incorporating this query-feedback into the state correction therefore makes state evolution better aligned with actual readout, rather than correcting memory only with respect to the write key.
>
> - The dissimilarity between the key and query is given as a supplementary to support the claim that key- and query-based readouts are not redundant and can provide complementary information from the same memory state.
>
> ### **W2. Performance Improvement**
>
> - We believe the improvement of Q-Delta is best interpreted in the context of the relative performance gaps among other baselines. As shown in Table 2, recent competitive models are themselves separated by relatively small margins, so even modest absolute gains can still be meaningful in this regime. From this perspective, Q-Delta achieves roughly 0.6\%p - 0.7\%p average improvement from second-best result at 340M and 1.3B scales, which is clear separation relative to the typical gaps among the 2nd--4th best recent linear RNN baselines.
>
> - Regarding the loss curve, we kindly suggest viewing this as a qualitative reference to check stable optimization, rather than as a measure of final model quality. We also want to note that, in the late training regime, all curves become largely overlapping. Within that regime, Q-Delta still remains slightly lower compared to others, which we view as consistent with its downstream advantage.
>
> - To statistically assess the variability of Q-Delta, we additionally repeated the 340M experiment over three random seeds and report the mean and standard deviation. Due to limited time and resources, we currently provide this multi-seed result for Q-Delta and Gated DeltaNet, which is our primary baseline. We plan to include ohter baselines as well in the final version.
>     - Also, we corrected a reporting error in the Avg. column of Table 2. The per-task values and conclusions are unchanged, you can check multi-run results and corrections in detail at the link below.
>
>         - [Table 6. Multi-seed results on language modeling and zero-shot common-sense reasoning](https://anonymous.4open.science/r/ICML-2026-Discussion-2CD2/Table%20corrections%20+%20Table%206.pdf)
>
>
> ### **W3. $\lambda$ and Performance**
>
> - [Table 8. Extended ablation results](https://anonymous.4open.science/r/ICML-2026-Discussion-2CD2/Table%208.pdf)
>
> - Extended ablation results show that the default learnable $\lambda_t$ gives the best overall performance. Fixed scalar choices underperform the learnable setting, while the intermediate value 0.5 gives the best result (47.20) among scalar variants ($\lambda = 0.2, 0.5, 0.8$).
>
>     - We also provide the distribution of $\lambda_t$ throughout pretraining on 340M Q-Delta.
>         - [Figure 5. Distribution of $\lambda_t$](https://anonymous.4open.science/r/ICML-2026-Discussion-2CD2/Additional_Figures.pdf)
>
>     - We observe that $\lambda_t$ is broadly distributed with a mean around 0.38, consistently taking nontrivial values across layers rather than collapsing toward zero. Having a broad, non-degenerate distribution, this implies that the model is actively using query feedback, but in a selective, input-dependent way.
>
> - We further performed ablation on the state decay/ forget factor $\alpha_t$ to validate the improvement from adding query-conditioned feedback to various types of delta-rules.
>     - Removing the decay factor ($\alpha_t = 1.0$) degrades performance (45.86 in Avg. accuracy), but it remains clearly above its most direct non-decay baseline, DeltaNet (43.82 in Avg. accuracy) at the same 340M scale.
>
> ### **Others**
>
> - Typo : We sincerely thank the reviewer for careful reading and letting us know the typos we've missed. We'll make sure to reflect these in our revised manuscript.
> - $\hat{o_t}$ simply corresponds to $S_{t-1} q_t$.

---

> > ### Author Rebuttal · Reviewer_gT2T · 2026-04-03
> >
> > 1. Eq. (16) depends only on $q_t$ and not on $q_\tau,(\tau<t)$, so why does it necessary to encode $q_t$ into $S_t$?
> >
> > 2. Q-Delta (which becomes GatedDeltaNet when $\lambda=0$) is a generalization of GatedDeltaNet. It has more parameters than GatedDeltaNet, so it performs better. The experiments should ensure that the number of learnable parameters is the same for both architectures, rather than treating Q-Delta as a direct generalization.

---

> > > ### Author Response · Authors · 2026-04-04
> > >
> > > ### **1. Query involvement in state**
> > >
> > > - We thank the reviewer for this clarification question. Eq. (16) characterizes the current query-conditioned readout from the prior state $S_{t-1}$, and it is not itself an update equation. The role of $q_t$ in Q-Delta is to define a query-conditioned prediction at step $t$, and then write its corrective signal into $S_t$ so that this information can influence future state evolution, rather than being used only as a transient probe.
> > >
> > > - Moreover, in the Q-Delta recurrence, **Eq. (16) depends not only on the current $q_t$, but also on past queries implicitly through the time-evolved keys $\tilde{k}_{\tau,t}$, since the state transition matrices contain $x_j = k_j + \lambda_j q_j$ aggregated across all previous time steps**. Thus, incorporating $q_t$ into the update is precisely what allows query-conditioned information to persist in memory and affect subsequent readouts.
> > >
> > > ### **2. Additional parameters**
> > >
> > > - Q-Delta adds only a lightweight query-feedback modulation $\lambda_t$, implemented by a small linear projection that gives headwise query-feedback coefficients from the hidden state. This introduces neither a new recurrent state nor an additional high-dimensional q,k,v transformation, but only a low-cost headwise controls for the strength of query feedback. As a result, the added parameter count is minimal and imposes essentially no practical architectural burden.
> > >
> > > - In addition, our scalar-$\lambda$ ablations provide a direct comparison for this. **With fixed scalar $\lambda$ (0.2, 0.5, 0.8), there are no extra learnable $\lambda_t$ parameters, yet the model still remains stronger than the $\lambda=0$ setting that recovers the GatedDeltaNet-style update.** Thus, the gain is not explained solely by parameter count and the query-conditioned correction itself is beneficial, while learnable $\lambda_t$ provides an additional benefit by adapting its strength.

---

### Official Review · Reviewer_Luzg · 2026-03-02

**Soundness:** 3
**Presentation:** 3
**Significance:** 3
**Originality:** 3
**Overall Recommendation:** 5
**Confidence:** 3

**Summary:**

The paper presents Q-Delta, which extends GatedDeltaNet by incorporating a query-induced correction term into the state update $S_t$. The authors build upon established Delta-rules where the state update $S_t$ introduces a damping of the term $v_{t}^{\text{old}} = S_{t-1} k_t$, representing the retrieved value the key $k_t$ would produce under the previous state $S_{t-1}$.


In analogy to this, the authors analyze the readout of the state $S_{t-1}q_t$  and propose a query-aware delta rule that adds a correction term linear in $ S_{t-1}q_t $. This proposed update rule is analyzed in detail both theoretically and experimentally. Across various benchmarks, Q-Delta is shown to outperform related architectures in most cases.

**Compliance With Llm Reviewing Policy:**

Affirmed.

**Final Justification:**

After a carefully reading of the answers I decide to keep my score because it reflects the value of this work.

**Key Questions For Authors:**

1. Would it be feasible to extend Theorem 3.2 and show that the prediction error for Q-Delta is smaller than for GatedDeltaNet, etc.?
2. Is there any explaination why the 340M model shows mixed benchmark results?

**Limitations:**

yes

**Strengths And Weaknesses:**

## Strengths
* The proposal to incorporate $S_{t-1}q_t$ into the delta rule is novel and is derived from a clear mathematical symmetry.
* The paper is well-written and thorough, providing sound mathematical theorems alongside a comprehensive benchmark comparison.
* The Q-Delta mixed error is shown to be mathematically stable and uniformly bounded.
* Benchmark results on 340M and 1.3B models are promising; notably, for the 1.3B model family, Q-Delta excels across a variety of evaluation tasks.

## Weaknesses
* While $S_{t-1}q_t$ is analyzed in mathematical detail, the paper would benefit from a more compelling intuitive explanation/ heuristic derivation for why this specific term should be included in the update rule. The section "Why query readout matters" is less convincing than the rest of the work:
    * The argument that Equation (16) can be expressed as a linear combination of value vectors is an expansion of $S_{t-1}$ into the (non-orthogonal) basis $v_\tau$, a property that holds for any vector $q_t$.
    * The empirical study demonstrating that $S_{t-1}q_t$ and $S_{t-1}k_t$ are statistically uncorrelated is expected.
    * An analysis analogous to the standard delta rule (suppression of redundant updates that would otherwise accumulate) using properties of the readout $S_{t}q_t$ would provide a better motivation for the method.
* Equation (16) was previously derived (in less generality) in "Gated Delta Networks: Improving Mamba 2 with Delta Rule" (page 2). This connection should be made more explicit.
* While the proposed method is promising in general, the empirical results for the 340M model are mixed across several benchmarks.

---

> ### Author Rebuttal · Authors · 2026-03-31
>
> We sincerely thank reviewer Luzg for the careful reading and valuable feedbacks, which help us sharpen both the presentation and the technical positioning of our work.
>
> ### **W1. Motivation**
>
> - We agree that the current presentation can better distinguish between what Eq. (16) shows algebraically and what actually motivates the update rule. As the reviewer notes, we agree that the identity $S_{t-1} q_t = \sum_{\tau < t} \langle q_t, \tilde{k_{\tau,t} \rangle_{\eta_\tau}} v_\tau$ is algebraically true for any arbitrary probe vector and is not by itself sufficient to justify incorporating query feedback into the update.
>
> - The key point we want to emphasize is that $q_t$ is not an arbitrary vector, it is the actual readout direction used to produce the downstream output, $o_t = S_t q_t$. Therefore, $S_{t-1} q_t$ is not merely another informative projection of the state, but the model's own value prediction along the very direction by which memory is ultimately consumed.
>
> - This gives a direct delta-rule style intuition, if the current memory already predicts part of the target value along the query direction, then rewriting that part is redundant. Incorporating this query-conditioned prediction into the correction therefore makes the update better focus on what's still missing, while aligning state evolution with the actual readout mechanism.
>
> ### **Q1. Comparison of prediction error dynamics**
>
> - Although a theorem showing pointwise direct error comparison over Gated DeltaNet is beyond our current analysis, Theorem 3.2 makes an important comparison point. Q-Delta corrects a joint key-query prediction error through the mixed corrective target $x_t = k_t + \lambda_t q_t$, whereas Gated DeltaNet focuses only on the key-conditioned prediction error. In this sense, Q-Delta uses a strictly richer corrective target than standard key-only delta-rule updates.
>
> - To verify this empirically, we add a figure that compares the error convergence dynamics between Q-Delta and Gated DeltaNet.
>     - [Figure 6. Value prediction error trace throughout training on 340 models](https://anonymous.4open.science/r/ICML-2026-Discussion-2CD2/Additional_Figures.pdf)
>     - As shown in Figure 6, Q-Delta exhibits more stable and bounded error tracking and converges to lower residual error not only for the query-based prediction $\|v-Sq\|$, but also for the key-based prediction $\|v-Sk\|$ and the mixed prediction $\|v-Sx\|$.
>     - In contrast, GatedDeltaNet converges to higher residual levels overall, and its dynamics remain much less sensitive to the query-conditioned component.
>
>
> ### **W2. Explicit link to Gated DeltaNet**
>
> - We appreciate the reviewer for pointing this out. We will make this connection explicit in our revised manuscript. The relation in Eq. (16) generalizes the query-readout expression revealed by Gated DeltaNet to a more general class of linear state transition operators $P_t$.
>
> ### **W3./Q2. On the 340M-scale results**
>
> - We view the 340M results as a smaller-scale setting in which the benefit of improved state correction appears less uniformly across tasks. Stronger results at larger scale suggest that the advantage of Q-Delta becomes clearer when the richer corrective dynamics can be more fully utilized.
> - To assess whether the reported result is stable, we additionally performed searches over three random seeds and report the mean and standard deviation.
>     - Also, we corrected a reporting error in the Avg. column of Table 2. The per-task values and conclusions are unchanged, you can check multi-run results and corrections in detail at the link below.
>
>         - [Table 6. Multi-seed results on language modeling and zero-shot common-sense reasoning](https://anonymous.4open.science/r/ICML-2026-Discussion-2CD2/Table%20corrections%20+%20Table%206.pdf)

---

> > ### Author Rebuttal · Reviewer_Luzg · 2026-04-01
> >
> > Thanks a lot for the comprehensive rebuttal, it answers my questions. As for the score, I'll keep 5: Accept.

---

> > > ### Author Response · Authors · 2026-04-01
> > >
> > > We're glad that our response addressed your concerns. Thank you for carefully reading our rebuttal and your thoughtful acknowledgement.

---

### Official Review · Reviewer_AJzU · 2026-03-13

**Soundness:** 2
**Presentation:** 3
**Significance:** 2
**Originality:** 2
**Overall Recommendation:** 3
**Confidence:** 3

**Summary:**

The paper addresses a meaningful modeling question: whether query information should influence recurrent memory evolution rather than only readout, and this motivation is presented clearly and consistently. The main technical contribution is well defined, combining a mixed key/query correction rule with a practical parallel training formulation and a stability discussion. The empirical section is reasonably broad and the reported results are favorable, especially on recall-heavy retrieval settings and average comparisons across benchmarks. At the same time, the experimental scale remains moderate, uncertainty estimates are missing, and several important ablations and reproducibility details are limited in the main text. The theoretical section is helpful for intuition, but its guarantees depend on empirical conditions rather than a sharper characterization of when those conditions should be expected to hold.

**Compliance With Llm Reviewing Policy:**

Affirmed.

**Key Questions For Authors:**

N/A

**Limitations:**

yes

**Strengths And Weaknesses:**

## Strengths

1. Clear and well scoped motivation: The paper identifies a concrete limitation in prior linear recurrent models: queries influence memory readout but not the evolution of the memory state.

2. Useful theoretical perspective: The manuscript openly states that the update is not strict gradient descent on a standard mixed objective, which is a helpful clarification. The stability discussion provides a structured way to reason about contraction and tracking behavior. The theory is also connected back to empirical measurements, which is better than presenting the analysis in isolation.

3. Well-established connection between modeling and implementation: The paper does not stop at a sequential recurrence, but also derives a parallel form suitable for efficient training. This is important because recurrent-style updates are only practically valuable if they remain compatible with modern accelerator-friendly training. The efficiency discussion and throughput results support the claim that the method preserves a competitive systems profile.

## Weaknesses
1. Experimental scale is still somewhat limited: The training setups are informative, but 1.3B model size and 15B training token is relatively small. It's better to provide a tendency of model performance as the model scales, so that we know if the advantage over baseline is actually getting bigger, or dimish as the model scales up.

2. Evaluation: The long-context evidence is still limited. While the paper reports synthetic retrieval results beyond the training context, the S-NIAH evaluation only extends to 4K tokens, which makes it difficult to judge whether the observed gains would persist in substantially longer and more practically relevant context regimes. Also the empirical results would be more convincing with uncertainty estimates, especially when some gains are moderate in magnitude (Table 2).

3. Ablation: Beyond the query-feedback coefficient, the paper provides limited component-level ablation for the rest of the design. Since Q-Delta combines several coupled choices—including the mixed key/query correction, gating, and the specific recurrent formulation—it would be useful to isolate which ingredients are most responsible for the gains, and whether the benefit mainly comes from query feedback itself or from interactions among these design choices.

4. The novelty: Although the paper presents a clean and useful extension, the conceptual step beyond prior delta-rule models may be narrower than the presentation suggests. Q-Delta still operates in the same linear associative-memory paradigm and modifies the corrective term by injecting query-conditioned feedback into state evolution. This is a sensible and empirically promising design, but it may be interpreted more as an extension of existing delta-style recurrent updates than as a clear break from prior formulations.

---

> ### Author Rebuttal · Authors · 2026-03-31
>
> We appreciate reviewer AJzU for careful reading and raising the constructive feedbacks, which help us sharpen both the presentation and the technical positioning of our work.
>
> ### **W1. Experimental scale**
> - We would like to first clarify our training setup, the 340M models were pretrained on 15B tokens, while the 1.3B models were pretrained on 30B tokens, under matched optimization settings.
> - We agree that evaluating larger scales is valuable future work. At the same time, we believe our current setup still provides a fair comparison with the existing line of works based on linear attention and SSMs. To our knowledge, prior works have typically reported results at scales no greater than 1.5B. Evaluated on two scales, 340M and 1.3B, Q-Delta shows a fair and effective improvement over existing linear RNN baselines.
>
> ### **W2. Evaluation**
>
> 1. **Longer context evaluation**
>     - We have conducted additional S-NIAH experiments at 8K context length, extending beyond the original 4K setting reported in Table 3.
>         - [Table 7. Retrieval performance on  S-NIAH benchmark on context length 8K](https://anonymous.4open.science/r/ICML-2026-Discussion-2CD2/Table%207.pdf)
>         - The results show Q-Delta continues to demonstrate stronger advantages in longer contexts, showing that the stable context evolution of Q-Delta persists under harsher condition.
>
> 2. **Uncertainty estimates**
>
>     - To assess the stability of the reported performance under randomness, we repeated the 340M experiment over three random seeds and report the mean and std. Due to limited time, we currently provide these multi-seed results for Q-Delta and Gated DeltaNet, which is our primary baseline. We plan to include other baselines as well in the final version.
>     - Also, we corrected a reporting error in the Avg. column of Table 2. The per-task values and conclusions are unchanged, you can check multi-run results and corrections in detail at the link below.
>         - [Table 6. Multi-seed results on language modeling and zero-shot common-sense reasoning](https://anonymous.4open.science/r/ICML-2026-Discussion-2CD2/Table%20corrections%20+%20Table%206.pdf)
>
>
> ### **W3. Ablation**
>
> - Given Q-Delta's recurrent formulation, $S_t = \alpha_t S_{t-1}(1 - \beta_t\,(k_t + \lambda_t q_t)\,k_t^\top) + \beta_t v_t k_t^\top$, the  only term that directly controls the strength of the query-feedback is $\lambda_t$. Since our main contribution is introduction of query-based state correction, our main ablation target is therefore $\lambda_t$. We also add ablation on the state decay/forget factor $\alpha_t$.
>     - [Table 8. Extended ablation results](https://anonymous.4open.science/r/ICML-2026-Discussion-2CD2/Table%208.pdf)
>
>     - Table 8 shows that the default learnable $\lambda_t$ gives the best overall performance. Fixed scalar choices underperform the learnable setting, while the intermediate value 0.5 gives the best result (47.20) among scalar variants ($\lambda = 0.2, 0.5, 0.8$). We also provide the actual distribution of $\lambda_t$ throughout training.
>         - [Figure 5. Distribution of $\lambda_t$](https://anonymous.4open.science/r/ICML-2026-Discussion-2CD2/Additional_Figures.pdf)
>
>     - Although removing the decay factor ($\alpha_t = 1.0$) degrades performance (45.86 in Avg. accuracy), it remains clearly above its most direct non-decay baseline, DeltaNet (43.82 in Avg. accuracy) at the same 340M scale.
>
> ### **W4. Novelty**
>
> - We thank the reviewer for raising this important point, which helps us clarify the precise scope and contribution of Q-Delta. We agree that Q-Delta remains within the broader delta-rule based linear associative-memory framework, and we do not intend to claim a wholesale departure from that family. Rather Q-Delta is a minimal yet conceptually nontrivial extension of the prior delta-rule.
>
> - Q-Delta changes a core assumption of conventional key-value associative paradigm, state updates are governed only by key-conditioned error while query treated as passive readout. However, query is precisely the direction producing the final output passed to downstream computations. Q-Delta incorporates this query-conditioned prediction into state evolution, making correction better aligned with actual readout. Not just heuristically adding query-feedback into state update, we further verify that introducing query feedback enables stable error dynamics under joint corrective signal combining both key and query-driven value prediction.
>
> - We further provide an empirical results that show Q-Delta actually achieves more stable and well-behaved error dynamics for both key and query based prediction compared to Gated DeltaNet throughout training.
>     - [Figure 6. Value prediction error trace throughout training on 340 models](https://anonymous.4open.science/r/ICML-2026-Discussion-2CD2/Additional_Figures.pdf)
>     - This suggests the proposed query-conditioned correction induces a genuine change in memory evolution dynamics.

---

> > ### Author Rebuttal · Reviewer_AJzU · 2026-04-05
> >
> > Thanks for the response, which partially addressed my concerns.
> > But the gain is still limited, I'll keep my score.

---

> > > ### Author Response · Authors · 2026-04-05
> > >
> > > - We appreciate the reviewer’s follow-up and the acknowledgment that our rebuttal addressed the concerns raised in initial review. For clarity, our understanding is that the remaining concern is less about an unresolved methodological issue and more about how to interpret the empirical gains of Q-Delta. On this point, we would like to emphasize the overall summary of results.
> > >
> > > - #### **Summary of performance comparison results. Zero-shot language modeling, commonsense reasoning average     accuracy, and S-NIAH long-context retrieval task (1K-4K average).**
> > >   | Model (1.3B)   | Wiki ppl. ↓ | LAMBADA ppl. ↓ | Avg. Reasoning ↑ | Avg. Retrieval (S-NIAH) ↑ |
> > >   | -------------- | ----------: | -------------: | ---------------: | ------------------------: |
> > >   | RetNet         |       22.45 |          21.84 |            50.31 |                     38.09 |
> > >   | Mamba          |       19.89 |          16.98 |            52.39 |                     62.62 |
> > >   | Mamba2         |       19.47 |          17.40 |            52.46 |                     76.58 |
> > >   | DeltaNet       |       19.77 |          16.64 |            52.53 |                     81.29 |
> > >   | Gated DeltaNet |       19.61 |          15.32 |            52.77 |                     83.51 |
> > >   | **Q-Delta**    |   **19.21** |      **15.19** |        **53.47** |                 **90.02** |
> > >
> > > - At the 1.3B scale, which is the largest setting in this paper, Q-Delta shows the strongest results across all four summary axes simultaneously. In particular, it achieves the best language modeling results on both reported metrics (e.g., WikiText and LAMBADA perplexity), the best average reasoning accuracy, and the best average S-NIAH retrieval score across 1K-4K setups (we further verify it on the 8K setup during rebuttal).
> > >
> > > - Taken together, this is not an isolated gain on a single task, but rather shows general trend across multiple tasks. Relative to the strongest prior delta-rule baseline, Gated DeltaNet, Q-Delta improves average reasoning accuracy by +0.70 points and long-context retrieval by +6.51 points, while also achieving lower perplexity on both language-modeling metrics. We therefore view the empirical gains of Q-Delta as a consistent and practically meaningful improvement across multiple capacities, rather than as a narrow or task-specific advantage.

---

### Decision · Program_Chairs · 2026-04-30

**Decision:**

Accept (regular)

**Comment:**

This paper proposes Q-Delta, a query-aware extension of the delta rule for linear-attention / linear-RNN models. Whereas standard delta-rule updates correct the state only against a key-conditioned value prediction, Q-Delta injects a query-conditioned prediction error into state evolution, so that the corrective signal aligns with the direction along which memory is actually read out. The paper provides a stability analysis, derives a chunkwise-parallel form with a Triton implementation that achieves throughput comparable to existing delta-rule baselines, and reports consistent gains on language modeling, zero-shot reasoning, and S-NIAH retrieval at 340M and 1.3B scales.

Reviewers consistently praised the clarity of motivation, the principled and minimal nature of the modification, the inclusion of stability theory and a hardware-efficient implementation, and the strong 1.3B-scale results, particularly on long-context retrieval. The author response further clarified the distinction from GatedDeltaNet (the modification is not merely an extra parameter, fixed-scalar λ ablations also outperform the GDN-equivalent λ=0 setting), added 8K-context S-NIAH results, multi-seed runs, and additional ablations on the decay factor and λ.

The main remaining concerns are that experimental scale is moderate (largest setting 1.3B / 30B tokens), the conceptual step beyond prior delta-rule models is narrow, and 340M-scale gains are less uniform across tasks than at 1.3B. I share the reviewers' overall view that, despite these limitations, the contribution is technically sound, well-motivated, and useful to the growing body of work on linear-attention architectures.

Overall, I recommend acceptance.